



# Evaluation of aerosol and cloud properties in three climate models using MODIS observations and its corresponding COSP simulator, and their application in aerosol-cloud interaction

Giulia Saponaro[1], Moa K. Sporre[2], David Neubauer[3], Harri Kokkola[1],
Pekka Kolmonen[1], Larisa Sogacheva[1], Antti Arola[1], Gerrit de Leeuw[1],
Inger H.H. Karset[2], Ari Laaksonen[1], and Ulrike Lohmann[3]

[1]Finnish Meteorological Institute, P.O. Box 503 FI-00101 Helsinki
[2]Department of Geosciences, University of Oslo, Norway
[3]Institute for Atmospheric and Climate Science, ETH Zurich, Zurich, 8092,
Switzerland

*Correspondence to:* Giulia Saponaro (giulia.saponaro@fmi.fi)

**Abstract.**

The evaluation of modeling diagnostics with appropriate observations is an important task that establishes the capabilities and reliability of models. In this study we compare aerosol and cloud properties obtained from three different climate models

5   ECHAM-HAM, ECHAM-HAM-SALSA, and NorESM with satellite observations using MOderate Resolution Imaging Spectrometer (MODIS) data. The simulator MODIS-COSP version 1.4 was implemented into the climate models to obtain MODIS-like cloud diagnostics, thus enabling model to model and model to satellite comparisons. Cloud droplet number concentrations (CDNC) are derived identically from MODIS-

10   COSP simulated and MODIS-retrieved values of cloud optical depth and effective radius. For CDNC, the models capture the observed spatial distribution of higher values typically found near the coasts, downwind of the major continents, and lower values over the remote ocean and land areas. However, the COSP-simulated CDNC values are higher than those observed, whilst the direct model CDNC output is significantly

15   lower than the MODIS-COSP diagnostics. NorESM produces large spatial biases for ice cloud properties and thick clouds over land. Despite having identical cloud modules, ECHAM-HAM and ECHAM-HAM-SALSA diverge in their representation of spatial and vertical distribution of clouds. From the spatial distributions of aerosol optical depth (AOD) and aerosol index (AI), we find that NorESM shows large biases





for AOD over bright land surfaces, while discrepancies between ECHAM-HAM and ECHAM-HAM-SALSA can be observed mainly over oceans. Overall, the AIs from the different models are in good agreement globally, with higher negative biases on the Northern Hemisphere. We computed the aerosol-cloud interactions as the sensitivity

of dln(CDNC)/dln(AI) on a global scale. However, one year of data may be considered not enough to assess the similarity or dissimilarities of the models due to large temporal variability in cloud properties. This study shows how simulators facilitate the evaluation of cloud properties and expose model deficiencies which are necessary steps to further improve the parametrization in climate models.

## 10  1   Introduction

A climate model is a powerful tool for investigating the response of the climate system to various forcings, enabling climate forecasts on seasonal to decadal time scales, and therefore can be used for estimating projections of the future climate over the coming centuries based on future greenhouse gas and aerosol forcing scenarios (Flato,

2011). Based on physical principles, climate models reproduce many key aspects of the observed climate and primarily aid to understand the dynamics of the physical components of the climate systems.

The evaluation of modeling diagnostics is an important task that establishes the capabilities and reliability of models. When key properties of the atmosphere (e.g.,

clouds, aerosols) are considered, the model assessment is relevant to assure that the climate model correctly captures key features of the climate system. The interest in the reliability of climate models reaches outside the scientific community, as these simulations will form the basis for future climate assessments and negotiations. Therefore, understanding the level of reliability is a necessary step to strengthen the robustness

of climate projections and, if necessary, improve the model parametrizations for the relevant processes.

For the evaluation of parametrizations of aerosol indirect effects in global models, satellite data have been proven to be useful (Quaas et al., 2009; Boucher et al., 2013) as they provide large spatial coverage at suitable temporal resolution. Satellite

instruments measure the intensity of radiation coming from a particular direction in a selected wavelength range. From the observed radiances, the geophysical quantities are then inferred by inverse modeling using a retrieval algorithm.





The compensation of modeling errors, the intrinsic uncertainties of observational data, and the possible discrepant definitions of variables between models and observational data are major issues affecting the crucial task of model evaluation. For that, satellite simulators have been developed to mimic the retrieval of observational data and to avoid ambiguities in the definition of variables mentioned above. Simulators recreate what the satellite would retrieve when observing the modeled atmosphere. By reprocessing model fields using radiative transfer calculations, they generate physical quantities fully consistent with the satellite retrievals. By including microphysical assumptions, which usually differ between models, inconsistencies in the simulators are avoided. Hence, simulators represent a robust and consistent approach not only for the application of satellite data to evaluate models, but also for model-to-model comparisons. Simulators have been widely used, and their implementation in several models enables intercomparison studies on atmospheric variables, such as clouds, aerosols (Quaas et al., 2009; Williams and Bodas-Salcedo, 2017; Zhang et al., 2010; Luo et al., 2017), and upper atmospheric humidity (Bodas-Salcedo et al., 2011).

Two prominent examples of simulators are the International Satellite Cloud Climatology Project, ISCCP, (Klein and Webb, 2009; Yu et al., 1996) and the CFMIP (Cloud Feedback Model Intercomparison Project) Observation Simulator Package, COSP (Bodas-Salcedo et al., 2011). CFMIP is part of The Coupled Model Intercomparison Project (CMIP) (Eyring et al., 2016b; Webb et al., 2017), which is a framework providing the modeling community with guidelines for the development, tuning and evaluation of models (Eyring et al., 2016a, c). COSP is a software tool developed within the CFMIP (Webb et al., 2017) which extracts parameters for several spaceborne active (CALIOP, CPR) and passive (MISR, MODIS) sensors.

In this study the COSP version 1.4 was implemented in three climate models, namely ECHAM-HAM, ECHAM-HAM-SALSA and NorESM, and the diagnostic outputs of the MODIS simulator were compared to MODIS observational data collected during the year 2008. The main goal of this study is to evaluate the models' capability to realistically represent clouds by employing MODIS satellite observations and its corresponding COSP simulator. A secondary goal of the study is to estimate the aerosol-cloud interaction (ACI) through the use of cloud droplet number concentration (CDNC) derived from observed and COSP simulated values of cloud optical thickness and effective radius. Also known as the first aerosol indirect effect (AIE) or sensitivity, the ACI is as an indicator ratio defined as the change in an observable cloud property (e.g., cloud optical depth, cloud effective radius, cloud droplet number





concentration) to a change in a cloud condensation nuclei proxy (e.g. aerosol optical depth, aerosol index, or aerosol particle number concentration). Originally introduced by Twomey (1977), the topic of ACI is still a major uncertainty in understanding climate change (e.g. Lohmann et al., 2007; Quaas et al., 2009; Storelvmo, 2012; Flato

et al., 2013; Lee et al., 2016)). The analysis of aerosol-cloud interaction has been reported in literature by a variety of methods: studies presenting results from global scales (Feingold et al., 2001; Quaas et al., 2010) to regional scales (e.g. Saponaro et al., 2017; Ban-Weiss et al., 2014; Liu et al., 2017, 2018) and in-situ observations (e.g. Sporre et al., 2014), using different approaches, i.e. observations from satellites,

airborne and ground based instrumentation, or modelling.

The choice of observations and spatial scale of a study presents intrinsic uncertainties when quantifying aerosol-cloud interactions, and some of them relate to spatial or temporal limitations or artifacts (McComiskey and Feingold, 2012). When considering satellite observations, cloud and aerosols properties are provided at a quite

comprehensive spatial and temporal coverage; however several aspects bring challenges in the analysis of these observations. The primary artifacts known to affect satellite estimation of aerosol-cloud interactions are related to (1) the inability of untangling aerosol and cloud retrievals from meteorology (e.g. aerosol humidification, entrainment, cloud regimes dependency), (2) inaccuracies in the retrieval algorithms

(e.g. twilight zone, contamination, statistical aggregation) and (3) assumptions in the retrieval algorithms (Koren et al., 2007; Oreopoulos et al., 2017; Christensen et al., 2017; Wen et al., 2007).

In this work, the Cloud Feedback Model Intercomparison Project (CFMIP) Observation Simulator Package (COSP) (Bodas-Salcedo et al., 2011) is implemented in

three climate models to obtain satellite-like diagnostics that enable a direct comparison with satellite retrieval fields. In particular, we focus on liquid cloud properties, which are used to derived CDNC. Cloud droplet number concentration is computed for both satellite observations and satellite-simulated values in a consistent way using an algorithm presented in Bennartz (2007). Aerosol-cloud interactions (ACI) are

quantified by dln(CDNC)/dln(AI). By considering the changes in CDNC, it is possible to isolate the microphysical component of the ACI without the need for constraining the liquid water path.

In Section 2 we provide details of the MODIS data, the models, and the COSP simulator. Section 3 presents the methods used in the analysis of the data. The evaluation

of the simulator cloud diagnostics with MODIS satellite data on a global scale is pre-





sented in subsections 4.1 and 4.2, while the ACI results are shown in subsection 4.3. Conclusions are summarised in Section 5.

## 2 Data

### 2.1 MODIS

The Moderate Resolution Imaging Spectrometer (MODIS) is a 36-channel radiometer flying aboard the Terra and Aqua platforms since 2000 and 2002, respectively, which views the entire Earth's surface every 1 to 2 days, thus representing an extensive data set of global Earth observations. MODIS delivers a wide range of atmospheric products including aerosol properties, water vapour, cloud properties, and atmospheric sta-
bility variables.

  We consider data for the year 2008 from MODIS-Aqua since its equatorial crossing time (13:30 local time) ensures a more complete development of the cloud during its daily cycle. MODIS Level-1 (L1) products are geo-located brightness and temperature values, which are elaborated into geophysical data products at Level-2 (L2), and ag-
gregated onto a uniform space-time grid at Level-3 (L3). We used the latest Collection 6.1 daily MODIS/Aqua MYD08L3, which is a regular gridded Level-3 daily global product (Hubanks et al., 2016). It contains daily 1° x 1° gridded average values of atmospheric aerosols properties and cloud optical and physical properties, along with a suite of statistical quantities, which are derived from the corresponding L2 atmo-
sphere data product. The dataset is limited to observations made during daytime, as these contain a richer set of retrievals and better accuracy in cloud detection.

  The Level-2 MODIS aerosol products provide information regarding the aerosol loading and aerosol properties over cloud-, snow-, and ice-free land and ocean surfaces at a spatial resolution of 10 km x 10 km. The primary aerosol product is the
aerosol optical depth (AOD), derived globally at the wavelength of 550 nm, while the other parameters accounting for the aerosol size distribution, such as the Ångström exponent (AE) or fine-mode aerosol optical depth, are only derived over ocean (Levy et al., 2013). Additionally, the aerosol index (AI) can be derived by multiplying AOD by AE. The MODIS aerosol products have been extensively validated using highly-
accurate observations made by the Aerosol Robotic Network (AERONET) (Sayer et al., 2014) showing good agreement with in-situ measurements. The uncertainty in MODIS retrievals of AOD from validation studies (Levy et al., 2007) was quantified at $0.03 + 0.05 \times \tau_A$ over ocean and $0.05 + 0.15 \times \tau_A$ over land, where $\tau_A$ is the refer-





ence AOD value from AERONET. In this study we primarily focus on the analysis of
liquid cloud properties. However, MODIS aerosol data (Levy et al., 2013) is needed
to assess aerosol-cloud interactions.

The Level-2 MODIS physical and optical cloud properties are derived trough a
combination of infrared emission and shortwave reflectance techniques at a spatial
resolution varying from 1 km to 5 km, depending on the parameter (Platnick et al.,
2017). Collection 6.1, which is used in this work, provides cloud optical parameters
divided into different products accordingly to the cloud phase and retrieved at wave-
lengths of 2.1 $\mu$m, at 1.6 $\mu$m and 3.7 $\mu$m (Hubanks et al., 2016; Platnick et al., 2017).
As the COSP simulator simulates cloud properties at 2.1 $\mu$m, the same wavelength
is selected in the MODIS observations for both ice and liquid clouds. MODIS offers
two scientific L3 cloud fractions datasets, namely the cloud mask cloud fraction and
the cloud optical properties cloud fraction (datasets with prefix 'Cloud Fraction' and
'Cloud Retrieval Fraction', respectively). From now on we refer to the cloud mask
cloud fraction as CF, and to the cloud optical properties cloud fraction as COP CF.
While the CF counts the proportion of the pixels classified by the cloud mask as
cloudy or partly cloudy, the COP CF counts the proportion of the pixels for which
cloud optical properties have been successfully derived. The main difference between
these two definitions roots in the approach of handling partly cloudy pixels. As the
task of the cloud mask is to identify fully clear pixels, partly cloudy pixels are counted
as cloudy in CF, while in the COP CF they are counted as clear because the retrieval
algorithm aims to include only fully cloudy pixels. The different treatment of partly
cloudy pixels directly impacts the number of cloud pixels, and consequently many
other retrieved cloud properties. Therefore differences are expected in our results and
as already reported by Pincus et al. (2012). MODIS observations are here used as a ref-
erence dataset. However, MODIS data contains its own errors and limitations. Many
studies compared MODIS liquid cloud microphysical properties with in-situ and air-
borne campaign measurements finding strong correlations for COT but a systematic
significant overestimation of MODIS cloud-top droplet effective radius (CER) for ma-
rine stratus and stratus cumulus clouds due to possible instrument limitation and al-
gorithm retrieval assumptions (e.g. Noble and Hudson, 2015; Painemal and Zuidema,
2011; Min et al., 2012). A good CER correlation between MODIS and in-situ data was
however observed by e.g. Preißler et al. (2016) for marine warm stratiform clouds at
higher latitudes. A bias in MODIS CER is propagated into the derivation of MODIS
LWP, which also shows a positive bias with respect to the observations (e.g. King





et al., 2013; Noble and Hudson, 2015; Painemal and Zuidema, 2011; Min et al., 2012). Overestimated MODIS LWP were also found over a high-latitude measurement land site (e.g. Sporre et al., 2016) for clouds from all altitudes in the atmosphere. Marchant et al. (2016) showed that the C6 cloud phase discrimination algorithm is significantly improved over C5 but some situations continue to be problematic over regions located at higher latitudes (i.e., polar areas, Greenland, and large desert areas).

In this study, we derive CDNC following the method presented in Bennartz (2007) and this additional cloud parameter is used in the computation of ACI. More information is provided in Sect. 3.2.

## 2.2 COSP - The CFMIP Observation Software Package

The simulator COSP (Bodas-Salcedo et al., 2011) is a publicly available software package (https://www.earthsystemcog.org/projects/cfmip/) developed by the CMIP community (Webb et al., 2017). It consists of a module coded in FORTRAN90 which simulates cloud properties and can be implement in any model.

The simulator's working principle is based on using climate model fields to mimic radiances to which a retrieval algorithm is applied to obtain satellite-like fields for the comparison with satellite observations.

This process is summed up in three main phases. As model grids are very coarse (∼100 km), the model fields are first down-scaled: each model gridbox mean profile is broken into subcolumns, whose size is more representative of a satellite retrieval area (∼10 km). Next, each sub-column profile is processed by a forward radiative transfer model to create synthetic radiances at the satellite retrieval area-level. The last step aggregates the simulator outputs to produce diagnostics (for example temporal averages and histograms) statistically comparable to the real satellite observations. A comprehensive explanation about the methodology and results of the COSP MODIS simulator is presented in Pincus et al. (2012).

## 2.3 Models

### 2.3.1 ECHAM-HAM

ECHAM-HAMMOZ (echam6.3-ham2.3-moz1.0) is a global aerosol-chemistry climate model (Schultz et al., 2018; Kokkola et al., 2018; Tegen et al., 2019; Neubauer et al., 2019) where ECHAM refers to the atmospheric model of the model configuration, HAM to the aerosol model, and MOZ to the chemistry model. In this study only





the global aerosol-climate model part of ECHAM-HAMMOZ is used. Instead of the comprehensive MOZ chemistry model, sulphate chemistry is calculated in HAM for which the details have been given by Zhang et al. (2012) and references therein.

ECHAM-HAMMOZ, referred to as ECHAM-HAM, consists of the general circu-
lation model ECHAM (Stevens et al., 2013) coupled to the latest version of the aerosol module HAM (Tegen et al., 2019) and uses a two-moment cloud microphysics scheme that includes prognostic equations for the cloud droplet and ice crystal number concentrations as well as cloud water and cloud ice (Lohmann and Diehl, 2006; Lohmann et al., 2007, 2008; Lohmann and Hoose, 2009).

Next to the two-moment cloud microphysics scheme the stratiform cloud scheme includes an empirical cloud cover scheme (Sundqvist et al., 1989).

The cirrus scheme is based on Kärcher and Lohmann (2002) and described in Lohmann et al. (2008), cloud droplet activation uses the Abdul-Razzak and Ghan (2000) parameterization, the autoconversion of cloud droplets to rain follows the method
from Khairoutdinov and Kogan (2000), immersion and contact freezing in mixed-phase clouds follows the scheme from Lohmann and Diehl (2006), and cumulus convection is represented by the parameterization of Tiedtke (1989) with modifications developed by Nordeng for deep convection.

Simulations were performed at T63 ($1.9° \times 1.9°$) spatial resolution using 31 verti-
cal levels (L31) and COSP v1.4. Horizontal winds and surface pressure were nudged towards the ERA-Interim (Dee et al., 2011) reanalysis for 2008, and observed sea surface temperatures and sea ice cover for 2008 were used (Taylor et al., 2000). Three-hourly instantaneous output was used. The COSP output is almost instantaneous as it is the three hour average over two hour time steps i.e. 50% of the values are instanta-
neous and the other 50% are an average over two time steps.

### 2.3.2 ECHAM-HAM-SALSA

ECHAM-HAM-SALSA is identical to the ECHAM-HAM setup (echam6.3-ham2.3-moz1.0), with the difference that the sectional aerosol module SALSA (Kokkola et al., 2008, 2018) is used instead of the modal model M7 used in the ECHAM-HAM
setup. SALSA calculates the aerosol microphysical processes: nucleation, coagulation, condensation, and hydration. In this setup, the aerosol model HAM applies also the sectional scheme for the rest of the aerosol processes, i.e. emissions, removal, aerosol radiative properties, and aerosol-cloud interactions. In addition to differences in the aerosol size distribution scheme, also the wet deposition schemes differ between





the ECHAM-HAM and ECHAM-HAM-SALSA setups. In addition, while ECHAM-HAM uses the cloud activation parameterization for modal models (Abdul-Razzak and Ghan, 2000), SALSA uses the activation parameterization for the sectional representation (Abdul-Razzak and Ghan, 2002). Along with the details of these differences, the

implementation and the evaluation of SALSA with the ECHAM-HAMMOZ model version which is used in this study has been presented by Kokkola et al. (2018).

Similarly to ECHAM-HAM, simulations were performed at T63 (1.9° × 1.9°) spatial resolution using 47 vertical levels (L47) and COSP v1.4. Horizontal winds and surface pressure were nudged towards the ERA-Interim (Dee et al., 2011) reanaly-

sis for 2008, and observed sea surface temperatures and sea ice cover for 2008 were used (http://www-pcmdi.llnl.gov/projects/amip/). Three-hourly instantaneous output was used.

### 2.3.3   NorESM

The Norwegian Earth System Model (NorESM) (Kirkevåg et al., 2013; Bentsen et al.,

2013; Iversen et al., 2013) is largely based on the Community Earth System Model (CESM) model (http://www.cesm.ucar.edu) but uses a different ocean model and a different aerosol scheme in the Community Atmospheric Model (CAM) (Neale et al., 2010).

The aerosol scheme in the NorESM version of CAM, called CAM-Oslo, can be

described as an aerosol life cycle scheme which calculates production tagged mass concentrations of different aerosol species (Kirkevåg et al., 2018).

In the current simulations, the NorESM model was run with the CAM-Oslo version 5.3 (Kirkevåg et al., 2018) which is configured with the microphysical two moment scheme MG1.5 (Morrison and Gettelman, 2008; Gettelman et al., 2015) for strati-

form clouds. The scheme includes prognostic equations for liquid (mass and number) and ice (mass and number) and a version of the Khairoutdinov and Kogan (2000) autoconversion scheme where subgrid variability of cloud water (Morrison and Gettelman, 2008) has been included. The aerosol activation into cloud droplets is based on Abdul-Razzak and Ghan (2000) and the heterogeneous freezing in CAM5.3-Oslo

is based on Wang et al. (2014) with a correction applied to the contact angle model (Kirkevåg et al., 2018). Moreover, CAM5.3-Oslo has a shallow convection scheme (Park and Bretherton, 2009) and a deep convection scheme (Zhang and McFarlane, 1995). The simulation was run with the Community Land Model (CLM) version 4.5 (Oleson et al.) with satellite phenology. Included in CLM is the Model of Emissions



of Gases and Aerosols from Nature (MEGAN) version 2.1 (Guenther et al., 2012) which interactively calculates the emissions of biogenic volatile organic vapors. Both isoprene and monoterpenes take part in the formation of secondary organic aerosol in CAM5.3-Oslo. The sea surface temperatures and sea ice in the simulation were prescribed monthly averages for the years 1982-2001.

The resolution for the simulation was $0.9° \times 1.25°$ and the surface pressures as well as horizontal winds were nudged against ERA-Interim reanalysis data (Berrisford et al., 2011) from 2008. CAM-Oslo was run with COSP version 1.4 producing three-hourly instantaneous outputs.

## 3 Methods

### 3.1 Post-processing of the datasets

The comparison of satellite retrievals and model variables is not always straightforward. Satellite-retrieved physical quantities may be derived slightly differently than the corresponding parameters in the model, and differences can be attributed to discrepancies in the retrieved quantities viewed from space versus model fields (i.e. retrieval assumptions, sensor limitations, spatial resolution) (Bodas-Salcedo et al., 2011). In this study we aim at highlighting the differences between observations and models which stem from different aerosol and cloud physical parametrization by using the COSP satellite simulator. Satellite simulators, such as COSP, represent a compromise between model fields and retrieved fields. Simulators use model fields to reproduce what the satellite sensor would see if the atmosphere had the clouds of a climate model. By taking the characteristics of the MODIS instrument into account, COSP generates simulated fields of cloud parameters which can be quantitatively compared to MODIS observations. The COSP diagnostics are then successively aggregated to the simulator outputs and are provided at the original model resolution. Prior to their intercomparison, post-processing of the COSP diagnostics and satellite data is necessary for obtaining a robust evaluation. COSP-derived parameters are in the original model resolution and represent grid-averaged values. As MODIS observations are grid values representative only of in-cloud pixels, the COSP grid-averaged values are divided by the corresponding cloud fractions. The three-hour outputs from the models were aggregated to daily averages and successively re-gridded and co-located by linear interpolation onto the finer satellite regular grid of $1°\times1°$. Each grid cell point of cloud variables from MODIS observations and MODIS diagnostics was screened



using a minimum threshold of 30% of cloud fraction to minimize the source of errors introduced by the retrieval algorithm and to ensure the existence of large-scale clouds. The screening does not introduce a significant loss in the data pool and provide grounds for a robust intercomparison as also shown in Bennartz (2007) and Ban-

Weiss et al. (2014). For each time step, only grid points having a valid observation simultaneously in each one of the four datasets were included in the final dataset for the statistical analysis.

The MODIS algorithm retrieves cloud properties in the proximity of the top of a cloud while the direct model outputs provide values through the entire vertical struc-

ture of a simulated atmospheric column. To overcome this issue, when comparing the direct model output CDNC and satellite-derived CDNC, for each grid box we selected the CDNC value at the top of the modeled cloud. Additionally, we selected only grid-points with temperature T > 273° K to exclude mixed-phase and ice clouds.

Note that all discussed cloud parameter are diagnosed using satellite simulators and

are compared to the corresponding MODIS satellite observations. However, we use two direct model diagnostics in the study:

- AOD, which is used to derive the AI, a proxy for cloud condensation nuclei for the computation of ACI

- $CDNC_{direct}$, which is compared with COSP-simulated and MODIS-derived es-

timates

### 3.2  Cloud droplet number concentration (CDNC)

The CDNC were derived from CER and COT from MODIS observations and COSP simulations by combining Eqs. (6) and (9) from Bennartz and Rausch (2017) in the following equation:

$$CDNC = \gamma \cdot COT^{0.5} \cdot CER^{-2.5},$$

where COT is cloud optical thickness, CER is the cloud droplet effective radius and $\gamma = 1.37 \cdot 10^{-5}$ m$^{0.5}$ (Quaas et al., 2006). The assumption of not accounting for temperature effect and setting $\gamma$ as a bulk costant applies rather well to the stratiform clouds in the marine boundary layer but less so for convective clouds (Bennartz, 2007;

Rausch et al., 2010).





### 3.3 Aerosol-cloud-interaction (ACI) computation

The aerosol-cloud-interaction (ACI) is defined here as the change in the selected cloud property, CDNC, to a change in AI, which is used here as a proxy for cloud condensation nuclei (CCN):

$$\text{ACI} = \frac{\text{dln(CDNC)}}{\text{dln(AI)}}$$

    The CDNC was computed from the CER and COT from the COSP-MODIS simulations and MODIS retrievals. Additionally, AI was derived from ECHAM-HAM, ECHAM-HAM-SALSA, and NorESM MODIS-COSP diagnostics, and MODIS satellite observations following Feingold et al. (2001). The mean values and standard de-

viations of the parameters involved in the computation of ACI are presented in Table 1. We discarded pixels retrieved when liquid cloud fraction is ≤0.3 to reduce noise-contamination and to focus on large-scale clouds. The screened parameters were used to derive CDNC.

    The ACI was calculated globally for each season. When computing ACI for large

areas, the ACI of each gridbox needs to be weighted by the corresponding number of data points (Grandey and Stier, 2010). This step was included in the post-processing of the datasets.

### 4   Results

#### 4.1   Global bias distributions

In this section we compare on a global scale aerosol and cloud properties from the three models by subtracting MODIS retrievals from the modeled COSP diagnostics. From now on we will refer to the difference between the simulated parameters and MODIS retrieved values using the term bias.

    Overall, the spatial distributions of the biases always show large discrepancies

around the polar and ice-covered areas, such as Greenland and Antarctica. Over these areas large discrepancies are expected due to the inaccuracy of the MODIS retrieval algorithm due to viewing geometry (i.e. large zenith or viewing angles) and to correctly classify opaque clouds, snow/ice surfaces and optically thin clouds over really bright or warm surfaces (Marchant et al., 2016).



Figure 1 presents the differences between the MODIS-COSP cloud fraction diagnostics and COP CF for ice clouds $CF_{ice}$ (Fig. 1b-d), and liquid clouds $CF_{liq}$ (Fig. 1f-h), as well as the differences between MODIS total COP CF (Fig. 1j-l), and CF (Fig. 1n-p). Additionally, for each comparison the MODIS spatial distribution is presented

as reference (Fig. 1a,e,i,m). It was already highlighted in section 2.1 that the cloud fraction retrieved from the optical properties ($CF_{ice}$, $CF_{liq}$ and COP CF) excludes partly cloudy pixels, representing a limitation in the comparison of the data. Thus, lower values of MODIS COP cloud fractions are expected. A widespread positive bias is observed for $CF_{ice}$ and $CF_{liq}$, indicating higher values of the COSP-simulated cloud

fractions than the MODIS observations. Prevalent cloud regimes can be recognized in the bias distributions. ECHAM-HAM and ECHAM-HAM-SALSA well represent the amount of ice clouds which are generally found in the intertropical convergence zone (ITCZ) and the marine subtropical stratocumulus and stratus regions, whereas liquid clouds are better represented over land areas and in the subtropical stratocumulus re-

gion. NorESM shows positive biases for ice cloud amount over stratus clouds regions and around the ITCZ, but shows smaller biases for liquid stratus cloud regimes than ECHAM-HAM and ECHAM-HAM-SALSA

The total cloud fraction bias shows a positive bias between the MODIS-COSP CF simulated by the three models and MODIS COP CF (Fig. 1j-l) and a negative bias

when MODIS CF is considered (Fig. 1n-p). Consequently, MODIS CF is higher than the MODIS COP CF product. This outcome is to be expected, and possibly originates from the different treatment in the MODIS algorithm of partly cloudy pixels in the computation of CF and COP CF, as discussed in section 2.1. Additionally, all models underestimate CF in marine subtropical stratocumulus regions.

The spatial distribution of the cloud physical and optical properties is remarkably similar among the datasets with the exception of $CER_{ice}$, IWP (Fig. 2 d and l) and COT (Fig. 3g,k) for NorESM. These strong biases are explained by the fact that in the NorESM COSP 1.4 implementation code includes radiative active snow in the computation of the effective radius and optical thickness of ice clouds. However, this

does not affect the properties of liquid clouds.

$CER_{ice}$ and IWP are underestimated in ECHAM-HAM and ECHAM-HAM-SALSA. This is likely caused by the cirrus scheme which does not account for heterogenous nucleation or pre-existing ice crystals during formation of cirrus clouds (Neubauer et al., 2019; Lohmann and Neubauer, 2018). Interestingly, dissimilarities can also be

observed between ECHAM-HAM and ECHAM-HAM-SALSA, despite the fact that





the models share the same cloud module. ECHAM-HAM $CER_{liq}$ is on average $5\mu$m smaller than in ECHAM-HAM-SALSA in the mid-latitude belt, and ECHAM-HAM-SALSA $CER_{liq}$ is larger around the polar areas (Fig. 2g) and shows a large positive bias for LWP over ocean (Fig. 2o) in comparison to ECHAM-HAM. LWP is also

overestimated by NorESM but over land areas (Fig. 2p), while ECHAM-HAM shows a good agreement with MODIS (Fig. 2n).

The evaluation of COT shows homogeneous results and comparable values of root mean square errors (Fig. 3) with the exception of NorESM COT biases for ice and liquid clouds which are particularly high over land. It appears that some tuning pa-

rameters, for example the autoconversion parameter, are particularly low and affect the convection scheme by suppressing precipitation, thus creating thick clouds. The comparison of the differences between the biases of ECHAM-HAM and of ECHAM-HAM-SALSA shows localized differences over India, China and Russia for IWP (Fig. 2j,k) and over China for water cloud COT (Fig. 3e,f). These are also regions

where aerosol microphysics has a fundamental role as shown in Kokkola et al. (2018). ECHAM-HAM and ECHAM-HAM-SALSA generally overestimate COT. The atmospheric model ECHAM shows a similar estimation when running without an aerosol model. This overestimation has been previouslt reported by Stevens et al. (2013).

Figure 4 shows global biases for CDNC derived from the MODIS retrievals and

the COSP diagnostics following the method presented in Sect.3.2 (Fig. 4b-d), and the daily averages of the direct output of the models (Fig. 4e-g). The differences between MODIS-COSP diagnostics and MODIS observations are very clear. Overall the MODIS derived CDNC is lower than that derived from COSP simulated values, but higher than the direct output values. Consequently, the CDNC from direct

model output is lower than MODIS-COSP diagnostics, as also found by Ban-Weiss et al. (2014). Possible explanations could be either related to the COSP method for deriving $CER_{liq}$ and $COT_{liq}$ or the approach used for deriving CDNC from $CER_{liq}$ and $COT_{liq}$. The biases between CDNC COSP-derived and modeled direct values are very different, but within each product the biases are similar, although local differ-

ences are observed. For example, the CDNC values from ECHAM-HAM-SALSA are lower in the polar regions and higher in the mid-latitude belt in comparison with the ECHAM-HAM and NorESM diagnostics. Local differences can also be observed in the direct output where ECHAM-HAM-SALSA shows higher values of CDNC over the oceans in the southern Hemisphere (Fig. 4f). A direct comparison of CDNC de-

rived from MODIS-COSP simulated variable and the model CDNC direct outputs is





shown in the supplementary material. ECHAM-HAM and ECHAM-HAM-SALSA were run with identical tuning parameter settings which were optimized for ECHAM-HAM. This choice was made to distinguish the differences in aerosol-cloud interactions coming from different aerosol microphysics modules. The differences in CDNC

between these two model setups originates from the cloud activation schemes, i.e. for HAM the modal cloud activation scheme of Abdul-Razzak and Ghan (2000) and for HAM-SALSA the sectional cloud activation scheme (Abdul-Razzak and Ghan, 2002). The cloud activation scheme of ECHAM-HAM-SALSA produces a higher number of CDNC than ECHAM-HAM (Fig. 4c) because SALSA microphysics module simu-

lates generally higher number of particles larger than 100 nm in diameter which act as cloud condensation nuclei. Despite the higher CDNC, $CER_{liq.}$ seems to be larger in ECHAM-HAM-SALSA than in ECHAM-HAM which is unexpected when assuming that both model version have similar LWC. This discrepant result may be explained by the fact that in ECHAM-HAM LWC is lower than in ECHAM-HAM-SALSA as a re-

sults of a systematically higher IWC. Thus, the $CER_{liq}$ diagnosed by ECHAM-HAM is also smaller despite of less CDNC. Differences in convective detrainment are likely linked with the result. In fact, a higher cloud droplet freezing rates are simulated in ECHAM-HAM-SALSA (except near the Equator) which could suggest reduced sedimentation of ice crystals less condensate being detrained as ice (and more as liquid)

in ECHAM-HAM-SALSA than ECHAM-HAM.

Figure 5 presents AOD and AI biases. The values of AI from direct model output and MODIS observations are quite close with an average bias of +0.2. The main divergence is observed in the ECHAM-HAM bias where higher AI values are simulated around the mid-latitude belt. Tegen et al. (2019) found indications that the particle

size of mineral dust and sea salt aerosol particles may be too small in ECHAM-HAM. More discrepancies can be observed in the AOD bias: ECHAM-HAM-SALSA AOD values are higher over ocean, and NorESM AOD are much higher over deserts and other bright surfaces (such as Africa and Australia). Other localized distinctions in aerosol loading distribution can be observed over regions which are typically strongly

affected by primary emissions (such as the Sahara, India, Southeast Asia, Russia, Canada, central Africa, and South America). The different representation of size distribution, microphysical processing of aerosols and sink processes has a significant effect on the modelled AOD as shown for the aerosol module SALSA2.0 by Kokkola et al. (2018). The overestimation of AOD in the tropical oceans and underestimation





of AOD at higher latitudes and over land in ECHAM-HAM has also been found by
Tegen et al. (2019).

## 4.2 Joint histogram

The analysis of the CTP-COT joint histogram enables to determine how well the data
sources represent the vertical cloud structures and regimes. Figure 6 shows the com-
parison of the simulated and observed global mean cloud fraction as a function of
cloud top pressure and cloud optical thickness. ECHAM-HAM and ECHAM-HAM-
SALSA (Fig. 6a,b) show a nearly identical result by concentrating a large fraction of
clouds at low level (CTP $\leq$ 680 hPa) and in the interval $3.6 \leq$ COT $\leq 23$. NorESM
(Fig. 6c) also concentrates its largest amount of clouds at low levels in the same COT
interval as in Fig. 6a and Fig. 6b, but detects also a higher fraction (about 2-2.5%) of
optically thick clouds $9.4 \leq$ COT $\leq 60$ throughout the atmosphere. A second cloud
fraction peak is observed for optically thin clouds (COT $\leq 1.3$) at very high levels
($180 \leq$ CTP $\leq 310$) for NorESM. This bimodal distribution resembles the vertical
distribution of the MODIS cloud fraction shown in Fig.6d. The MODIS observations
are mostly in the category of high-level clouds (CTP $\leq 440$ hPa) and low-level clouds
($680$ hPa $\leq$ CTP). MODIS shows on average more mid-level clouds than NorESM
and a higher fraction at low-level for $3.6 \leq$ COT $\leq 23$ similarly to ECHAM-HAM
and ECHAM-HAM-SALSA. Figure 6e shows the differences in cloud vertical distri-
bution where MODIS is generally having the highest cloud fraction except for mid-
level. MODIS also presents the highest percentage of clouds for COT $\geq 3.6$. NorESM
and MODIS detects nearly the same amount of clouds for $1.3 \leq$ COT $\leq 3.6$, while
for optically very thin clouds (COT $\leq 1.3$) a good agreement is obtained between all
datasets and NorESM shows the highest percentage of cloud fractions.

## 4.3 Aerosol-cloud interactions

The global daily mean values of CDNC and AI were used to assess how clouds are
affected by the changes of the CCN proxy. Uncertainties were computed as the 95%
confidence intervals using daily averages. Positive estimates of ACI indicate an in-
crease of CDNC as a function of AI, which could be an indication of the aerosol in-
direct effects. The potential limitations to this approach are further discussed in Sect.
5.

Figure 7 shows estimates of ACI on a global scale, including both land and ocean,
for each season and, separately, for the entire period under study as 'All'. The same





analysis is iterated on a regional scale and presented in the supplementary material
(Fig.S4) Error bars are representative of the boundaries of the 95% confidence inter-
val. ACI from the model results is generally positive suggesting that changes in AI are
connected with an increase of CDNC and the trend seems to be independent of the

time of the year. The modeling ACI estimates are similar in the models; however, the
results are statistically indistinguishable owing to fully overlapping confidence bars
(Cumming et al., 2007). MODIS ACI estimates show negative values for the win-
ter months (DJF), especially over the Northern Hemisphere (Fig.S4). As the global
estimates include land areas, these negative values could be indicative of retrieval bi-

ases over bright surfaces (i.e. snow or ice). Furthermore, negative ACI values may be
associated with the presence of different types of aerosol (i.e. hydrophobic aerosol
such as dust, black carbon) and their proximity to clouds, which may affect or in-
hibits the growth of cloud droplets (Chen et al.; Jiang et al., 2018; Costantino and
Bréon, 2013). Over ocean negative ACI values from MODIS observations have been

systematically found over subtropical marine stratuscumulus regions (i.e. N. Atlantic
Ocean, N.America, S.Atlantic Ocean). In these regions Chen et al. (2014) found a de-
crease in LWP with increasing AI for non-precipitating scenes. Additionally, negative
ACI values were suggested owing to wet scavenging or mixing of environmental air
by entrainment (Ackerman et al., 2004). While both processes affect LWP, CDNC is

not necessarily changing. This indicates limits in the derivation of CDNC from re-
trieved quantities for MODIS. Also water uptake by aerosol particles and effects of
meteorology can have a significant impact on the estimation of ACI derived from the
relationship between CDNC and AI (Neubauer et al., 2017).

Cloud properties (Fig. 2 and Fig. 3) are more similar for ECHAM-HAM and ECHAM-
HAM-SALSA, which share the same atmospheric model, rather than between the
two and NorESM. Nevertheless, the ACI estimates show good agreement between the
three models and, even more important, with ACI derived from MODIS observations.

## 5    Summary and Conclusions

The differences between observed and modeled aerosol and cloud properties can be
related to many factors, among which are the different parametrizations of aerosol
and cloud physical processes in the models, or differences in observation characteris-
tics by satellite, as well as meteorological influences on aerosol-cloud interactions. In
this study we focus on the differences due to the physical parametrization of aerosol





and cloud properties, and minimize the impact of the other factors. This objective was achieved by using a satellite simulator, which resolves the issue related to the incongruities between model and satellite views, and by nudging modeled winds to meteorological observation, solving the discrepancies between observed and modeled
meteorology.

The results show that the aerosol module in a climate model, in our case ECHAM-HAM and ECHAM-HAM-SALSA, has a smaller effect on the simulation of cloud properties than switching to another atmospheric model, NorESM. However, the three models differ from each other in the spatial and vertical representation of clouds. The
COSP cloud fraction diagnostics are comparable to MODIS products but the difference between the two MODIS products of total cloud fractions is significant. Despite having identical cloud modules, ECHAM-HAM and ECHAM-HAM-SALSA diverge when comparing liquid water cloud properties yet both fail to represent high level clouds. The discrepancies between ECHAM-HAM and ECHAM-HAM-SALSA
may originate from different amounts of activated droplets and different ice nucleation rates. While the NorESM cloud vertical distribution is closer to MODIS, large biases are found globally for cloud droplet size and water content in ice clouds due to the contribution of radiatively active snow (Kay et al., 2012). The inclusion of radiatively active snow in the physical model and the COSP module mitigates the underestimation
of model mid-level and high clouds but heavily impacts the magnitude of the global values of the cloud properties in ice clouds.

The differences observed in the simulation of cloud properties are reflected in the estimations of ACI. ACI is generally larger for ECHAM-HAM and NorESM, while being lower for ECHAM-HAM-SALSA and MODIS where the latter is the only
dataset leading to negative ACI values possibly owing to the linkages between aerosol and cloud type and their location in the atmosphere.

Although satellite simulators allow robust comparisons, their reliability is flawed when the observational data is not well explained or the simulator itself fails to address specific characteristics. Therefore, their strengths and weaknesses need to be
accounted for as to successfully use simulation diagnostics in model-observation comparisons as illustrated in details by Pincus et al. (2012), and Kay et al. (2012) to successfully use simulator diagnostics in model-observation comparisons. For example, simulators have limitations in depicting horizontally heterogeneous cloud regimes as they do not account for sub-pixel clouds which may explain the differences in the de-
tection of small cloud fractions between observations and models. However, simulator





and observational errors are here neglected because we considered them to be less important in the explanation of the model biases. The observed biases in the modeled clouds could originate from errors in the model calculation as well from the cloud parametrization; the identification of the specific reasons for these discrepancies is

beyond the scope of this study.

The results presented here indicate that the cloud droplet number concentration appears to be more sensitive to changes in aerosols in models than observations and these results are in agreement with many previous studies found in the literature (e.g. Ban-Weiss et al., 2014; Quaas et al., 2004; McComiskey and Feingold, 2012; Pen-

ner et al., 2011). Some of the differences in the ACI estimates from satellites and models could be associated with limitations in satellite measurements. For example, the estimates of ACI might suffer from an averaging effect due to the large spatial averages of satellite aerosol and cloud properties. L3 data can introduce spurious relationships between aerosols and cloud properties (e.g. McComiskey and Feingold,

2012; Christensen et al., 2017), and provide a rather limited pool of data samples enabling the analysis only over large regions. This was not explored in this study because we used the same spatial resolution for both the true model estimate and for the satellite-based model estimate for the ACI. Neubauer et al. (2017) performed a detailed study on the impact of meteorology, cloud regimes, aerosol swelling, and

wet scavenging on microphysical cloud properties using ECHAM-HAM. The results highlight that a minimum distance between cloud and aerosol gridded data should be taken into account, and that dry aerosols should be selected to reduce the influence of aerosol growth due to humidity. Similarly to our results, Neubauer et al. (2017) find a systematical overestimation of the sensitivity of modeled LWP and CDNC com-

pared to MODIS observations, and often a disagreement in sign in the comparison of cloud parameters. The results suggest that the derivation of CDNC from satellite observations may be limited by entrainment mixing of environmental air or precipitation. Furthermore, the models can not resolve the entrainment mixing at the top of stratocumulus clouds, which puts the LWP sensitivity to aerosol change in the mod-

els into question. In conclusion, this study identified limitations and deficiencies in the models, and their acknowledgment is important for the model development process and the correct interpretation of modelling diagnostics. We highlighted many discrepancies in cloud spatial and vertical representations and the results showed that the three models overall similarly represent the stratocumulus cloud regime being un-

derestimate when compared to MODIS. We discovered that IWC is systematically



lower in ECHAM-HAM-SALSA than in ECHAM-HAM due to a higher cloud droplet freezing rate which consecutively triggers a reduced sedimentation of ice clouds. This outcome explains the contradictory result in ECHAM-HAM-SALSA that shows the largest global averages for CER among the models despite having the highest number

of CDNC. Further investigation is needed to explain the differences in ice cloud properties between ECHAM-HAM and ECHAM-HAM-SALSA. The clouds simulated by NorESM are too thick over land and this issue is not only seen in COSP-variables but also in the default model output due to a very low autoconversion parameter which caused the suppression of precipitation over land, thus thicker clouds. Additionally,

in support to Ban-Weiss et al. (2014), the study revealed that the direct model CDNC is systematically larger than the values derived from COSP-diagnostics and MODIS observation.

Finally, we point out that the model deficiencies identified here may lead to an improvement of model parametrization and to more robust results. As future work,

a regional-based analysis would enable a better understanding of the physical processes responsible for the model biases. Additional research should be conducted to evaluate the aerosol-cloud-interaction following the approach suggested by Neubauer et al. (2017). These further steps would potentially benefit the modeling community interested in climate applications.

*Data availability.* The MODIS satellite data used in this study are publicly available at https://ladsweb.nascom.nasa.gov. ECHAM-HAM data are available from David Neubauer (david.neubauer@env.ethz.ch), ECHAM-HAM-SALSA data are available from Harri Kokkola (harri.kokkola@fmi.fi) and NorESM data from Moa K. Sporre (moa.sporre@nuclear.lu.se).

*Author contributions.* ECHAM-HAM-SALSA, ECHAM-HAM, and NorESM data (and corresponding descriptive text in Sect. 2.3 Data) were provided by DN, HK, and MS, respectively. IH assisted in setting up the NorESM simulations. GS conducted the data analysis, and wrote the majority of manuscript, except for the sections describing the models. PK, HK, AA participated in reading the results. GL, UL, PS helped to set up the concept idea of the paper. MS, DV, HK, PK, UL and GL contributed to review the manuscript.

*Competing interests.* No competing interests to declare.





*Acknowledgements.* The research leading to these results has received funding from the European Union's Seventh Framework Programme (FP7/2007-2013) Project BACCHUS under Grant Agreement 603445. We would like to thank Jennifer E. Kay for support in the implementation of COSP1.4 in NorESM. The ECHAM-HAMMOZ model is developed by a consortium

5   composed of ETH Zurich, Max Planck Institut für Meteorologie, Forschungszentrum Jülich, University of Oxford, the Finnish Meteorological Institute and the Leibniz Institute for Tropospheric Research, and managed by the Center for Climate Systems Modeling (C2SM) at ETH Zurich.





**Figure 1.** Annual global mean bias in cloud fraction. The bias represents the difference between MODIS-COSP diagnostics from ECHAM-HAM, ECHAM-HAM-SALSA, NorESM and MODIS observations. COSP-simulated total ice and liquid cloud fractions are compared with MODIS retrieval ice fraction (b-d), and with MODIS retrieval liquid cloud fraction (f-h), respectively. COSP-simulated total cloud fraction is compared with MODIS retrieval total cloud fraction (COP CF) (j-l), and cloud mask cloud fraction (CF) (n-p). Pixels with liquid cloud fraction $\leq 30\%$ are screened. The averages represent in-cloud values. High latitudes (Lat $> 60°$ N or Lat $> 60°$ S) are excluded in the computation of the root mean square error (RMSE). MODIS spatial distribution is presented as reference (a,e,i,m).





**Figure 2.** Annual global mean bias in cloud effective radius and water path. The bias represents the difference calculated subtracting MODIS observation to MODIS-COSP diagnostics from ECHAM-HAM, ECHAM-HAM-SALSA, and NorESM. Ice cloud effective radius ($CER_{ice}$) from MODIS-COSP is compared with MODIS observations in (b)-(d) and liquid cloud effective radius ($CER_{liq}$) in (f)-(h). The biases related to the comparison of COSP-simulated ice water path (IWP) are showed in (j)-(l) and for liquid water path (LWP) in (n)-(p). Pixels with liquid cloud fraction $\leq$ 30% are screened. The averages represent in-cloud values. Pixels with liquid cloud fraction $\leq$ 30% are screened. Values are in-cloud concentrations. High latitudes (Lat > 60° N or Lat > 60° S) are excluded in the computation of the root mean square error (RMSE). MODIS spatial distribution is presented as reference (a,e,i,m).



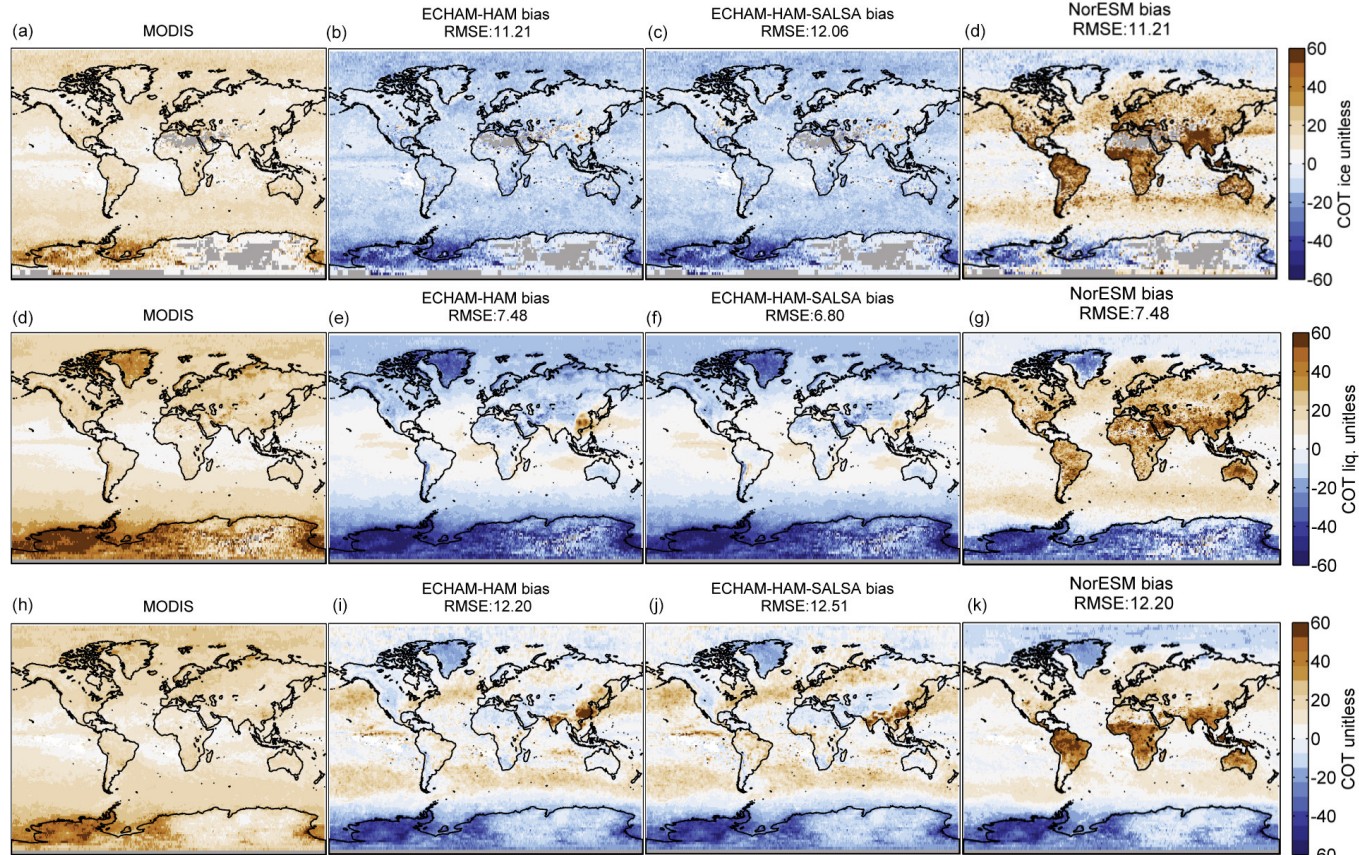

**Figure 3.** Annual global mean bias in cloud optical thickness for ice clouds (b-d), liquid water clouds (e-g) and total (combined ice and water clouds) COT (i-k) between MODIS and ECHAM-HAM, ECHAM-HAM-SALSA, and NorESM. The bias represents the difference between MODIS-COSP diagnostics and MODIS observations. Pixels with liquid cloud fraction $\leq 30\%$ are screened. The averages represent in-cloud values. High latitudes (Lat $> 60°$ N or Lat $> 60°$ S) are excluded in the computation of the root mean square error (RMSE). MODIS spatial distribution is presented as reference (a,d,h).

,





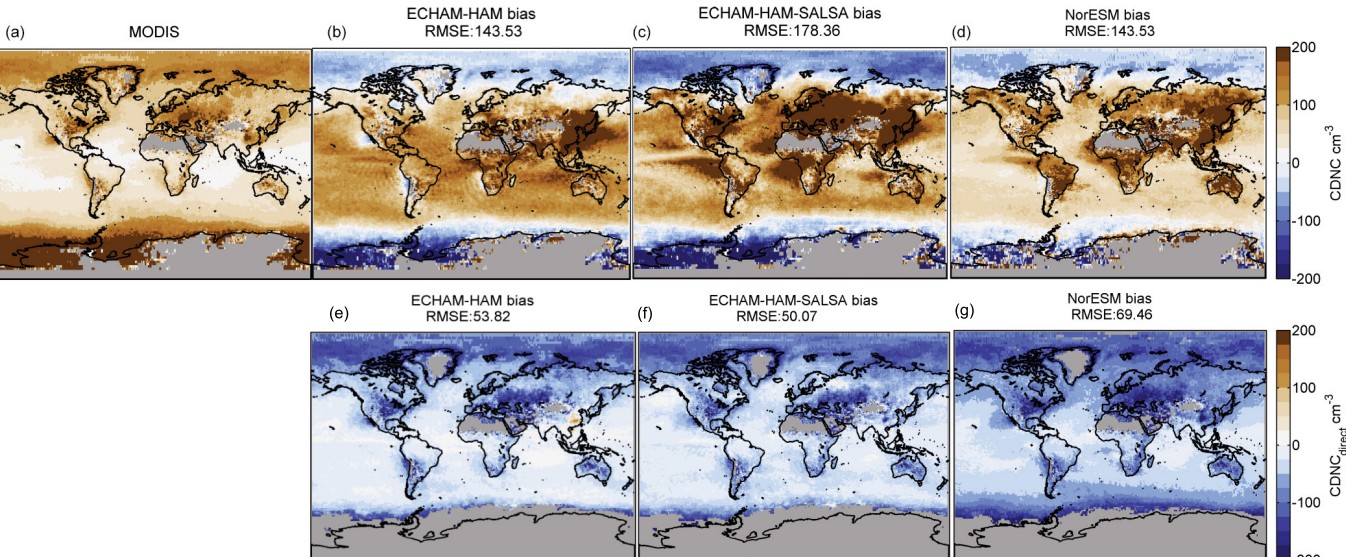

**Figure 4.** Cloud droplet number concentration (CDNC) annual mean bias.The bias represents the difference between CDNC derived from MODIS-COSP diagnostics and MODIS observations(b-d), and the model direct outputs and MODIS observations (f-h). Pixels with liquid cloud fraction $\leq 30\%$ are screened. The averages represent in-cloud values. High latitudes (Lat $> 60°$ N or Lat $> 60°$ S) are excluded in the computation of the root mean square error (RMSE). MODIS spatial distribution is presented as reference (a,e).



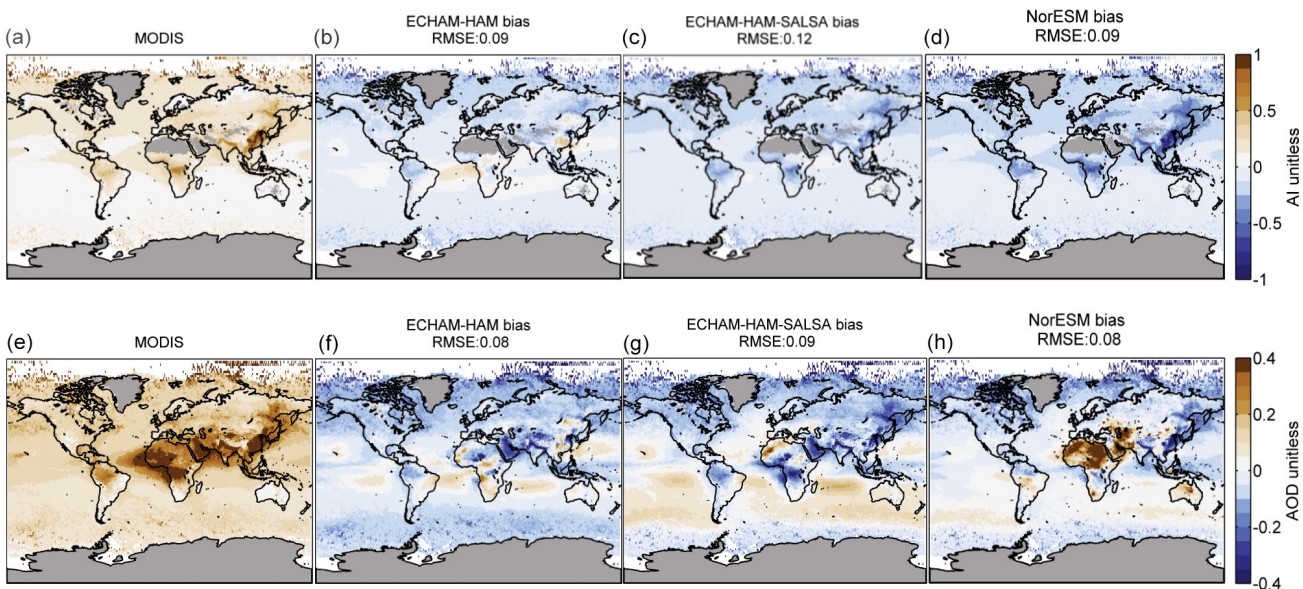

**Figure 5.** Aerosol Index (AI) (b-d) and Aerosol Optical Depth (f-h) annual mean bias. The bias represents the difference between the model direct outputs and MODIS observations. High latitudes (Lat > 60° N or Lat > 60° S) are excluded in the computation of the root mean square error (RMSE). MODIS spatial distribution is presented as reference (a,e).





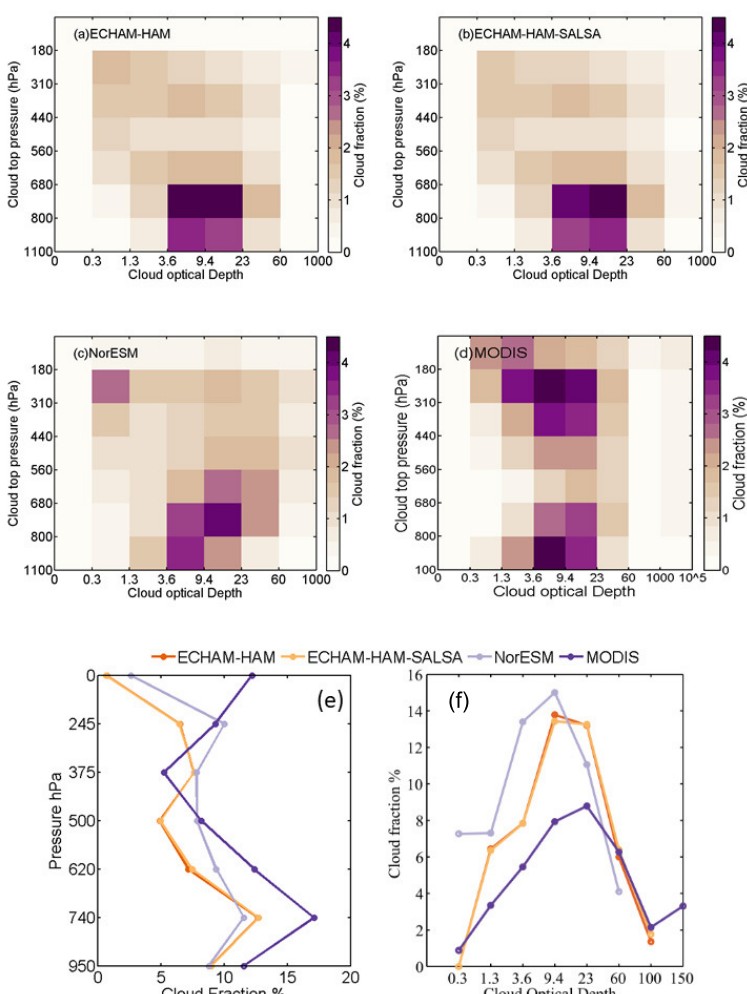

**Figure 6.** Vertical distribution analysis. Cloud fraction as a function of cloud top pressure and optical thickness for (a) ECHAM-HAM, (b) ECHAM-HAM-SALSA, (c) NorESM and (d) MODIS. The color scale represents the cloud fraction percentage. (e) Cloud fraction as a function of CTP (sum of all optical depth $\geq$0.3, and (f) cloud fraction as a function of COT (sum of all CTP layers for each COD-bin).





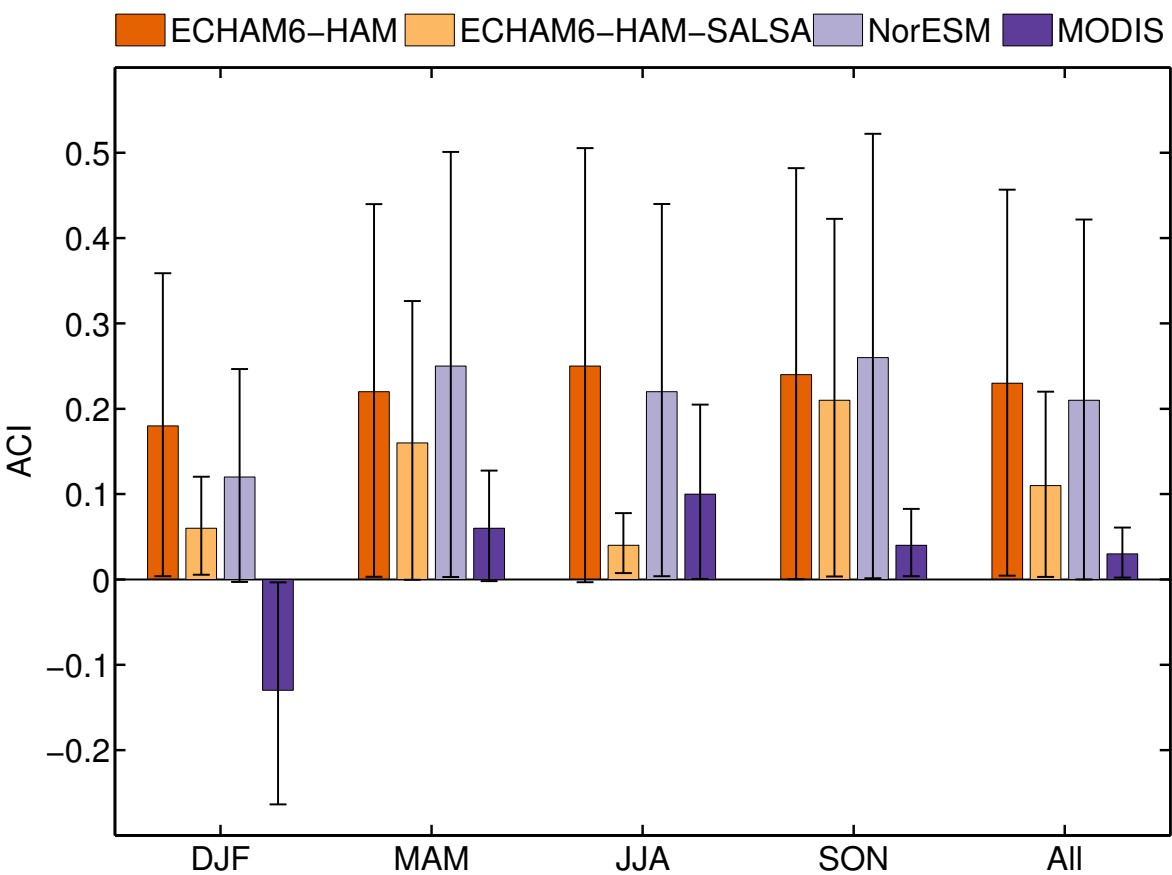

**Figure 7.** Global estimates of the aerosol-cloud interaction (ACI) computed as the changes of ln(CDNC) to ln(AI). CDNC are derived from corresponding daily grid points of LWP and COT from MODIS observations and COSP-MODIS outputs following Bennartz (2007). Global ACI values are calculated by season and for the entire period (1 January 2008 – 31 December 2008). Uncertainties estimates are calculated as 95% confidence interval from the daily values.





**Table 1.** Annual global in-cloud mean value $\pm$ standard deviation for the parameters used in the study. If a grid point has CF $\leq$ 30%, the point is set to fill values in all the datasets. The process leads to a reduction of 35% of datapoints in each dataset. 'CF all' is not screened for CF $\leq$ 30%.

| Source | CF all | CF | LWP $gm^{-2}$ | CER $\mu m$ | COT | CDNC $cm^{-3}$ | AI |
|---|---|---|---|---|---|---|---|
| MODIS | 0.68$\pm$ 0.35 | 0.83 $\pm$ 0.21 | 140 $\pm$ 142 | 15.3 $\pm$ 4.7 | 18.5$\pm$18.7 | 82 $\pm$ 82.12 | 0.15$\pm$0.20 |
| ECHAM-HAM | 0.56 $\pm$ 0.36 | 0.70 $\pm$ 0.21 | 106 $\pm$ 83 | 11 $\pm$ 1.9 | 9.6 $\pm$ 11.9 | 168 $\pm$ 122 | 0.14$\pm$0.20 |
| ECHAM-HAM-SALSA | 0.56 $\pm$ 0.36 | 0.70 $\pm$ 0.20 | 168 $\pm$ 159 | 12.5 $\pm$ 3.5 | 9.9$\pm$ 11.9 | 177 $\pm$ 183 | 0.11$\pm$0.18 |
| NorESM | 0.63 $\pm$ 0.34 | 0.42 $\pm$ 0.28 | 161 $\pm$ 133 | 11.9 $\pm$ 2.7 | 28.3 $\pm$ 53.6 | 167 $\pm$ 124 | 0.17 $\pm$ 0.26 |



**Table 2.** Summary of the the models used in the study.

| Model | Reference | Resolution | Aerosol scheme | Cloud microphysics |
|---|---|---|---|---|
| ECHAM-HAM | Tegen et al. (2019) | $1.9°$ lat $\times 1.9°$ lon, 31 levels | HAM 2.3-M7 | 2-moment scheme |
| ECHAM-HAM-SALSA | Kokkola et al. (2018) | $1.9°$ lat $\times 1.9°$ lon, 31 levels | HAM2.3-SALSA | 2-moment scheme |
| NorESM | Kirkevåg et al. (2018) | $0.9°$ lat $\times 1.25°$ lon, 30 levels | OsloAero | 2-moment scheme MG1.5 |



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
