# Peer review of "Evaluation of aerosol and cloud properties in three climate models using MODIS observations and its corresponding COSP simulator, and their application in aerosol-cloud interactions"

_Atmospheric Chemistry and Physics, 2019_

## Referee Comment (RC1) · Anonymous Referee #1 · 4 Oct 2019

This manuscript uses a satellite simulator to evaluate cloud and aerosol properties from 3 models against MODIS-satellite-derived properties. Overall, it is useful to employ these simulators to translate the model properties to those derived by the satellites. The paper is generally publishable for ACP, but have one major issue regarding the use of MODIS AI over land as well as a number of more minor issues.

Major comment:

The methods section admits that the MODIS Angstrom Exponent product (used in the

Aerosol Index [AI] calculation) is not calculated over land due to its low data quality in these locations. However, Figure 5 still shows MODIS AI values over land. Why is this?

In Levy et al., 2013, which describes the collection 6 Dark Target product, it says, "On a global basis, we and others have found little quantitative skill in MODIS-retrieved aerosol size parameters over land (e.g., Levy et al., 2010; Mielonen et al., 2011). We have decided to discontinue further attempts at validating Ångström Exponent (AE) and fine-AOD. A user can still choose to derive AE (from spectral AOD) or fine-AOD (from product of $\tau$ $\eta$) and evaluate the results themselves."

Levy et al. 2013: https://www.atmos-meas-tech.net/6/2989/2013/amt-6-2989-2013.pdf Levy et al. 2010: https://www.atmos-chem-phys.net/10/10399/2010/acp-10-10399-2010.pdf

So did you calculate AE from the spectral AOD over land? Is this any good? Is there any value to compare MODIS AI to the model's AI over land if MODIS AE over land does not have skill?

I think that the AI values over land should be removed from Figure 5 and discussion unless these values are tested against e.g. AERONET.

Specific comments:

P2 L16: I'd remove "primarily" here as climate models serve many purposes.

P3 L16-17: This sentence makes it seem like ISCCP is itself a cloud simulator. However, ISCCP is much broader than this, and foremost it has observational data products. Could say "are the simulator developed as part of the of International Satellite…"

P3 L23 and many places throughout: The clause following "which" is a non-restrictive clause (it does not help specify which simulator you're writing about and only provides additional information about it), which means there should be a comma before "which". If it were a restrictive clause, it would continue to not have a comma, e.g., "We use the cloud simulator which was developed as part of CMIP" (the clause af-

ter "which" is necessary to know the specific simulator you are referring to). I found many non-restrictive clauses throughout that did not contain commas, so please update. http://www.cws.illinois.edu/workshop/writers/restrictiveclauses/

P3 L24: Should these acronyms be defined here?

P3 L33: I'm generally used to AIEs specifically referring to the radiative effects of ACIs rather then being a synonym for ACIs in general, as AIE is presented here.

P4 L20: Although Koren et al., 2007 is cited at the end of the sentence, it seems jargony to list "twilight zone" without definition. May be more clear to replace with "near-cloud impacts on radiative transfer".

P5: I'm confused as to why the MODIS L2 products are discussed in detail after it is stated that L3 products were using in the paper. I assume it is because the L3 product is built from the L2 product (which is stated), but it would be good to make it clear why the L2 products are discussed in detail.

P5 L22-: Which aerosol product(s) are you using? Just the Dark Target product or also Deep Blue? I assume MAIAC is not used since it says the spatial resolution is 10x10 km. It looks like the Dark Target - Deep Blue combined product is used in Figure 5 based on there being AOD information over deserts etc.

P5 L27: Here it says that AE is only derived over ocean, which is correct, but why does AI have land values in Figure 5? See Major Comment above.

P7 L19: How are the model fields downscaled? It seems like there would be a lot of necessary assumptions to break a partially cloudy coarse gridbox into finer subcolumns. These assumptions should affect the results in theory. At a minimum, please add a statement such as, "details of this downscaling process and assumptions are provided in XX", assuming that this process has been documented elsewhere. If these details haven't been documented, please do so here or in the supplement.

P8 L4: "...referred to *here* as. . ."

P8 L23-25: This sentence didn't make sense to me. Please rewrite for clarity.

P8 L29: M7 should be mentioned/discussed in the previous subsection on ECHAM-HAM (without SALSA).

P9 L20: What does "aerosol life cycle scheme which calculations production tagged mass" mean? Is this an aerosol microphysics scheme? Does it track aerosol composition by its emission/process source in addition to chemical composition? Please rewrite for clarity.

P9 L34: Reference is missing a year.

P11 L22: "CER" isn't defined until below.

P12 L24-25: Is there a figure that we should be looking at to see these biases?

P13 L26: Do you specifically mean the *model* datasets here, or is the MODIS data being lumped into this comparison too.

P15 L22: It seems very subjective to say that a bias of -0.2 is "quite close" given that most of the globe has an AI below 0.2 according to MODIS (so this bias is larger than the AI value in nearly all locations. (Also, most of the locations with AI > 0.2 are over land, where we should not trust the Angstrom Exponent).

P17 L9: There is a discussion here about AI over land, but there is no acknowledge-ment that the MODIS aerosol team does not publish AE over land.

P18 L25-26: "possibly owing..." onward. It is unclear to me what this is saying.

P19 L3-4: What is the difference between "model calculation" and "cloud parameteri-zation". These seem like synonyms? Or is the "model calculation" specifically referring to the COSP simulator (rather than the atmospheric model).

P19 L22: How does one select dry aerosols when using satellite-derived properties? Or is this a statement of when using modelled properties only?

P19 L34-35: What property is being underestimated?

---

## Author Comment (AC1) · 18 Oct 2019

We would like to thank the referee for carefully reading our paper and for the helpful comments and suggestions. We have modified the manuscript according to these suggestions, and detailed answers to each comment are listed below. The reviewer comments are in italic and our answers are in normal font. In the modified manuscript the changes are shown in red font. The modified manuscript can be found from the supplement material of this post.

[Figure]
* * *
Interactive comment

*Major comment:*

*The methods section admits that the MODIS Angstrom Exponent product (used in the C1 ACPD Interactive comment Printer-friendly version Discussion paper Aerosol Index [AI] calculation) is not calculated over land due to its low data quality in these locations. However, Figure 5 still shows MODIS AI values over land. Why is this? In Levy et al., 2013, which describes the collection 6 Dark Target product, it says, "On a global basis, we and others have found little quantitative skill in MODIS-retrieved aerosol size parameters over land (e.g., Levy et al., 2010; Mielonen et al., 2011). We have decided to discontinue further attempts at validating Ångström Exponent (AE) and fine-AOD. A user can still choose to derive AE (from spectral AOD) or fine-AOD (from product of $\tau \eta$) and evaluate the results themselves."*
*Levy et al. 2013: https://www.atmos-meas-tech.net/6/2989/2013/amt-6-2989-2013.pdf*
*Levy et al. 2010: https://www.atmos-chem-phys.net/10/10399/2010/acp-10-10399-2010.pdf*
*So did you calculate AE from the spectral AOD over land? Is this any good? Is there any value to compare MODIS AI to the model's AI over land if MODIS AE over land does not have skill? I think that the AI values over land should be removed from Figure 5 and discussion unless these values are tested against e.g. AERONET.*
Author response:
We agree with the reviewer that the use of MODIS AI over land has not carefully been explained in the manuscript. The Ångström exponent over land was derived from spectral AOD. Although the resulting AI over land is shown in Figure 5, the values are not used for the calculation of mean values in Table1 not in the calculation of ACI (Figure 8). For these reasons, the author agrees with the reviewer: AI over land is masked out from Figure 5 and from the discussion section. Furthermore, the text now clearly states that AI excludes values over land.
Changes:
Figure 5 and Figure 7 were updated and replaced. The text describing the figures and

the results were updated throughout the manuscript.

**Specific comments:**

*P2 L16: I'd remove "primarily" here as climate models serve many purposes.*
Author response:
Accepted.

*P3 L16-17: This sentence makes it seem like ISCCP is itself a cloud simulator. However, ISCCP is much broader than this, and foremost it has observational data products. Could say "are the simulator developed as part of the of International Satellite. . ."*
Author response:
Accepted. Sentence modified as suggested.

*P3 L23 and many places throughout: The clause following "which" is a non-restrictive clause (it does not help specify which simulator you're writing about and only provides additional information about it), which means there should be a comma before "which". If it were a restrictive clause, it would continue to not have a comma, e.g., "We use the cloud simulator which was developed as part of CMIP" (the clause after "which" is necessary to know the specific simulator you are referring to). I found many non-restrictive clauses throughout that did not contain commas, so please update. http://www.cws.illinois.edu/workshop/writers/restrictiveclauses/*
Author response:
Clause updated at P7, L14; P9, L24; P10, L2, L19 and L24; P14, L9; P15, L18; P18, L34.

*P3 L24: Should these acronyms be defined here?*
Author response:
Accepted. Acronyms have been included and the sentence has been rephrased as: "COSP is a software tool developed within the CFMIP (Webb20 et al., 2017), which extracts parameters for several spaceborne active sensors, such as the Cloud-Aerosol Lidar with Orthogonal Polarization (CALIOP) and the Cloud Profiling Radar (CPR), as well as for passive sensors, such as the Multi-angle Imaging Spectro-Radiometer (MISR) and the Moderate Resolution Imaging Spectroradiometer(MODIS).

*P3 L33: I'm generally used to AIEs specifically referring to the radiative effects of ACIs rather then being a synonym for ACIs in general, as AIE is presented here.*
Author response:
The nomenclature describing the aerosol-cloud-radiation interaction has been changing throughout the years and the IPCC 5AR (IPCC2013) introduced the new terminology which is used in the manuscript.
Changes: the acronym AIE is not used and the abbreviation AIE has been removed from the text.

*P4 L20: Although Koren et al., 2007 is cited at the end of the sentence, it seems jargony to list "twilight zone" without definition. May be more clear to replace with "near-cloud impacts on radiative transfer".*
Author response:
Accepted. The sentence has been modified as suggested: "The primary artifacts known to affect satellite estimation of aerosol-cloud interactions are related to (1) the inability of untangling aerosol and cloud retrievals from meteorology (e.g. aerosol humidification, entrainment, cloud regimes dependency), (2) inaccuracies in the retrieval algorithms (e.g. near-cloud impacts on radiative transfer, contamination, statistical aggregation) and (3) assumptions in the retrieval algorithms (Koren et

al.,2007; Oreopoulos et al., 2017; Christensen et al., 2017; Wen et al., 2007)."

*P5: I'm confused as to why the MODIS L2 products are discussed in detail after it is stated that L3 products were using in the paper. I assume it is because the L3 product is built from the L2 product (which is stated), but it would be good to make it clear why the L2 products are discussed in detail.*
Author response:
The sentence was expanded to explain why MODIS L2 data are described. The new sentence is: "As the L3 1 x 1 gridded average values of atmospheric properties, along with a suite of statistical quantities, are derived from the corresponding L2 atmosphere data product, a brief description of Level-2 MODIS aerosol and cloud products is now presented."

*P5 L22-: Which aerosol product(s) are you using? Just the Dark Target product or also Deep Blue? I assume MAIAC is not used since it says the spatial resolution is 10x10km. It looks like the Dark Target - Deep Blue combined product is used in Figure 5 based on there being AOD information over deserts etc.*
Author response:
Indeed the combined product Dark Target - Deep Blue is used in this study.

*P5 L27: Here it says that AE is only derived over ocean, which is correct, but why does AI have land values in Figure 5? See Major Comment above.*
Author response:
See Author response to Major Comment.

*P7 L19: How are the model fields downscaled? It seems like there would be a lot of necessary assumptions to break a partially cloudy coarse gridbox into finer*

*subcolumns. These assumptions should affect the results in theory. At a minimum, please add a statement such as, "details of this downscaling process and assumptions are provided in XX", assuming that this process has been documented elsewhere. If these details haven't been documented, please do so here or in the supplement.*
Author response:
The following sentence citing the proper documentation has been added at P7, L25-L26: "A comprehensive explanation about the methodology and results of the COSP MODIS simulator is presented in Pincus et al. (2012)."

*P8 L4: "...referred to \*here\* as. . ."*
Author response:
Accepted.

*P8 L23-25: This sentence didn't make sense to me. Please rewrite for clarity.*
Author response:
The sentence was to explain that instantaneous output cannot be used with the implementation of the COSP satellite simulator in ECHAM-HAM. The sentence was rewritten to:
"The output of the COSP satellite simulator is also three-hourly. The implementation of the COSP satellite simulator in ECHAM-HAM does not allow instantaneous output. The COSP satellite simulator is called every radiation time step (i.e. every two hours) and the output of the COSP satellite simlator is averaged over the three hourly output period. This means that on average 50% of the values in the output of the COSP satellite simulator are instantenous values (i.e. from only one time step) and 50% of the values are an average over two radiation time steps (i.e. an average over two instantaneous values which are two hours apart)."

*P8 L29: M7 should be mentioned/discussed in the previous subsection on*

[Figure]

*ECHAMHAM (without SALSA).*
Author response:
A brief description of M7/HAM was added: "Aerosol microphysical processes such as nucleation, coagulation, condensational growth are computed by the modal scheme M7 (Vignati et al, 2004). HAM computes further processes such as emissions, sulfur chemistry (Feichter et al., 1996), dry depositon, wet deposition, sedimentation, aerosol optical properties, aerosol-radiation and aerosol-cloud interactions."

*P9 L20: What does "aerosol life cycle scheme which calculations production tagged mass" mean? Is this an aerosol microphysics scheme? Does it track aerosol composition by its emission/process source in addition to chemical composition? Please rewrite for clarity.*
Author response:
We have rewritten and expanded the description of the aerosol scheme to improve clarity as follows: "The aerosol microphysics scheme in the NorESM version of CAM, called CAM-Oslo, consists of 12 log-normally shaped background modes which are tagged accord-ing to emission source and chemical composition (Kirkevåg et al., 2018). The shapeof these modes can change due to condensation and coagulation."

*P9 L34: Reference is missing a year.*

*P11 L22: "CER" isn't defined until below.*
Author response:
The acronym CER is already introduced at P6, L34.

*P12 L24-25: Is there a figure that we should be looking at to see these biases?*
Author response:

We included the reference to the corresponding figures.

*P13 L26: Do you specifically mean the \*model\* datasets here, or is the MODIS data being lumped into this comparison too.*
Author response:
Indeed we referred to the model datasets. The sentence was modifed as suggested: "The spatial distribution of the cloud physical and optical properties is remarkably similar among the model datasets with the exception of $CER_{ice}$, IWP (Fig. 2 d and l)and COT (Fig. 3g,k)."

*P15 L22: It seems very subjective to say that a bias of -0.2 is "quite close" given that most of the globe has an AI below 0.2 according to MODIS (so this bias is larger than the AI value in nearly all locations. (Also, most of the locations with AI > 0.2 are over land, where we should not trust the Angstrom Exponent).*
Author response:
The sentence was unclear. The author meant that a similar bias is shown by the three models, as each bias is on average about 0.2. The sentence has been rephrased as: "The biases between values of AI from directmodel output and MODIS observations are quite close among the model as their average is about of +0.2."

*P17 L9: There is a discussion here about AI over land, but there is no acknowledgement that the MODIS aerosol team does not publish AE over land.*
Author response:
The following sentence was added: "As these negative values are derived over land regions, it could be indicative of retrieval biases over bright surfaces (i.e. snow or ice). Furthermore, it is important to inform the readers that MODIS aerosol size parameters over land (i.e. AE or fine-AOD) are no longer official products directly provided by the MODIS aerosol team. The publication of these variable were discontinued due to low

quantitative MODIS skill (Mielonen et al.,2011; Levy et al., 2013). Using spectral AOD, we derived AE over land and derived AI on a global scale to allow estimates of ACI on a global scale (Fig.S4). However, the AE values over land were not evaluated."

*P18 L25-26: "possibly owing. . ." onward. It is unclear to me what this is saying.*
Author response:
The sentence has been updated accordingly to the new version of Figure 7.

*P19 L3-4: What is the difference between "model calculation" and "cloud parameteri-*
*zation". These seem like synonyms? Or is the "model calculation" specifically referring*
*to the COSP simulator (rather than the atmospheric model).*
Author response: The terms "model calculation" and "model parametrization" describe different aspects of atmospheric modeling. While the term "model calculation" refers in the sentence to the COSP simulator, the term parametrization refers to the climate model. In particular, the latter term is used to describe the approach implemented in any atmospheric model to simplify too complex or too detailed processes to be explicitly resolved withing the model.

*P19 L22: How does one select dry aerosols when using satellite-derived properties?*
*Or is this a statement of when using modelled properties only?*
Author response:
The sentence refers to the study that Neubauer performed using the model ECHAM-HAM.
The sentence has been rephrased in the manuscript as follow: "The results highlight that a minimum distance between cloud and aerosol gridded data should be taken into account in the anaylsis of satellite data, and that dry aerosols should be selected to reduce the influence of aerosol growth due to humidity for model simulations when comparing satellite-based and model estimates for ACI."

[Figure]

*P19 L34-35: What property is being underestimated?*
Author response:

Cloud fraction is the parameter omitted in the sentence. The sentence has been rephrased as: "
[revised manuscript text omitted]

---

## Referee Comment (RC2) · Anonymous Referee #2 · 14 Nov 2019

In this work three different atmospheric models (ECHAM-HAM, ECHAM-HAM-SALSA and NorESM) are compared against MODIS retrievals by using the COSP instrument simulator. Emphasis is given on the estimation of cloud droplet number concentration (CDNC) and its sensitivity to aerosol load, as a way to assess the importance of aerosol-cloud interactions. The paper is well written and the subject is of relevance to the atmospheric community. In general the paper is correct and I don't see any major flaw that would limit publication. At the same time it is hard to tell what original contribution this paper offers, other than a throughout description of the models performance.

In that sense I think this work would be more suitable for a journal like Geophysical Model Development. If the authors still want to publish it in ACP a deeper exploration of the observed differences in CDNC-aerosol index sensitivity must be included. I have made some suggestions below.

Comments

The authors simply mention that the MODIS-derived CDNC and the direct model output are different, which is true, but offer no explanation of why those difference appear. Intuitively, the authors assume that the CDNC at cloud top should correspond to the MODIS retrieval, but omit the fact that MODIS does not produce CDNC directly. Instead using the method from Bennartz et al . (2007) one may argue that CDNC at cloud base or even the maximum CDNC in the column should be used. Also since the assumption of adiabaticity and vertically constant CDNC is embedded in the MODIS-derived CDNC the authors should limit the analysis only to regions where those assumption are valid (probably mostly over ocean).

The global CNDC-aerosol index sensitivity calculations (what the authors mistakenly call ACI throughout the manuscript) are also mostly descriptive and even though some speculation is given on the possible causes for discrepancy I imagine there is enough model output to go deeper (see specific comments). Also, in principle calculating the "ACI" from 2D vertically integrated fields makes little sense. It is not clear what role the assumptions of overlap are, or even whether the cloud and the aerosol occupy the same space.

Finally there is the issue of "the ACI". Aerosol-cloud interactions encompass many processes occurring in clouds as a result of the presence of aerosols. As a noun, aerosol-cloud interactions is an area of study, not a metric. So it is troubling, and in many places grammatically incorrect, that ACI are reduced to a single number and equated to the CDNC-AI sensitivity. Please correct this and be precise in the terminology used.

Specific Comments

Page 3, Line 34. Why 2008? The horizontal resolution is low enough that it should be easy to run a couple of years at least. MODIS data spans at least 15 years as well.

Page 3, Line 35. No, the aerosol- cloud interaction is not a metric, and certainly not confused with the aerosol indirect effect. Even saying "the aerosol cloud interaction" is probably incorrect. Moreover, ACI is referred here as a metric and later on as a topic. Please be precise in the terms used.

Page 4 Line 32. This is the CDNC-AI sensitivity. Also I am not convinced that the meteorological component is completely removed, please explain.

Page 4 Line 32. Why should the liquid water path remain constant? Is there a clear connection between the CDNC-AI sensitivity and the first aerosol indirect effect?

Page 9, Line 17. Why is the vertical resolution different than when running ECHAM-HAM?

Page 11, Line 6-8. This seems incorrect and could be a major flaw. COSP should account for the fact that MODIS only sees in-cloud values. It makes little sense to divide two column-integrated, 2D values. If COSP cannot account for it, then the 3D model calculation should be converted to in-cloud values before sending it to COSP to calculate MODIS-like values. Please clarify.

Page 11, Line 18. Is there a MODIS algorithm to retrieve CDNC? It may be more correct to say in MODIS cloud effective radius is biased towards cloud top values, which may propagate to the CDNC calculation. The method used to estimate CDNC (equation 1) should approximate better cloud-base values.

Page 12, Section 3.2. Please label this equation, and explain the assumptions behind it. Given than clouds are 3D, to what vertical level should this calculation correspond? Also, if this applies well in the stratiform marine boundary layer why are the authors applying it globally? What would be the error incurred in applying it to a shallow convective trade cumulus for example?

Page 12, Line 15. This is the CDNC-AI sensitivity, not "the ACI".

Page 12, Lines 16-28. Please expand this. Is this done using a linear regression of the CDNC and AI daily time series? Given that this is a global calculation why would the number of data points be different? Also, it is not clear what the "ACI" from 2D vertically integrated fields represents (see general comments).

Page 13, Line 18. Does COSP produce COP CF or CF all? Maybe the discussion should be limited to that one.

Page 15, Lines 1-10. This is very vague and should be explored in more detail (see comments above).

Page 15, Lines 17-20. Is the difference due to the activation scheme or to the aerosol models?

Page 15, Lines 25-30. These two sentences contradict each other.

Page 15, Lines 30-35. This is purely speculative.

Page 17, Line 15-25. Could this be corroborated? It seems odd that the sensitivity would be negative.

Page 17, Line 25. Global maps of CDNC-AI sensitivity should show this better.

---

## Author Comment (AC2) · 27 Nov 2019

We would like to thank the referee for carefully reading our paper and for the helpful comments and suggestions. We have modified the manuscript according to some of the suggestions and more detailed answers to each comment are listed below. The Reviewer's comments are in italic and our answers are in normal font. In the modified manuscript the changes (including those implemented after the comments from Referee 1) are shown in red font. The modified manuscript can be found from the supplement material of this post.

[Figure]

During the review of the manuscripts, two errors were found in the text:

1. We selected the maximum column value of CDNC of direct output of the models and not the values at cloud top. The text has been revised accordingly at page 11, line 29.

2. We refer in the text to AI, and consequently, ACI estimates, for land areas only. Clearly, this is incorrect as AI values are shown over ocean only. The text is revised:

   - at page, 4 line 35;
   - at page 5, line 1;
   - in the caption of Figure 5 and Figure 7;
   - in the caption of Table 1.

*General comment:*

*The authors simply mention that the MODIS-derived CDNC and the direct model output are different, which is true, but offer no explanation of why those difference appear. Intuitively, the authors assume that the CDNC at cloud top should correspond to the MODIS retrieval, but omit the fact that MODIS does not produce CDNC directly. Instead using the method from Bennartz et al. (2007) one may argue that CDNC at cloud base or even the maximum CDNC in the column should be used. Also since the assumption of adiabaticity and vertically constant CDNC is embedded in the MODIS-derived CDNC the authors should limit the analysis only to regions where those assumption are valid (probably mostly over ocean).*

Author response:

We present some possible explanation regarding the differences between MODIS-derived CDNC and the direct model output for CDNC (page 15, line 1-10). However, we do not enter in details as it would go beyond the scope of the paper. The result, being the direct modeled CDNC being lower than the COSP-derived CDNC, is supported by a similar conducted by Ban-Weiss et al. (2014). Therefore, this is an interest topic that should be investigated in future work.

The paragraph has been modified as:

"Consequently, the CDNC from direct model output is lower than MODIS-COSP diagnostics. Possible explanations could be either related to the COSP method for deriving $CER_{liq}$ and $COT_{liq}$ or the approach used for deriving CDNC from $CER_{liq}$ and $COT_{liq}$ or related to the fact that the computation of the direct CDNC requires a minimum number of CDNC set to 40 $cm^{-3}$. This outcome is very similar to what has been found by Ban-Weiss et al. (2014), where the CDNC satellite-simulated values are higher than the standard CAM5 model output near clout top as well as the column maximum value. Therefore, this result represent an interesting topic that extends beyond the scope of this paper but it should further developed in future work. "

Further discussion on the different results between the model direct CDNC output and the COSP-derived CDNC is included in section 4.1, page 15, lines 4-20. Considerations regarding the limitations of MODIS-derived CDNC are presented in the manuscript. Alternative methodologies exists for deriving CDNC at cloud base such as the approach developed by Rosenfeld et al. (2016). However, this method is specifically design for Suomi NPP satellite retrievals and based on adiabacity assumptions. Ban-Weiss et al. (2014) showed that the modelled CDNC is lower than values derived from satellite observation regardless the selection of CDNC at cloud top or the maximum CDNC in the column.

MODIS provides three separate products of the cloud properties (i.e. cloud effective radius, optical depth and water path) using three different water absorbing channels (1.6 $\mu$m, 2.1 $\mu$m and 3.6 $\mu$m). The signal of the these three wavelengths implies different penetration depths in the cloud system, which has clearly obvious consequences

on the capabilities of the wavelengths to be responsive to cloud top or near-cloud-top microphysics as shown in Rosenfeld et al. (2003), for example. The 3.6 $\mu$m has the least in cloud penetration depth, therefore the most representative of cloud top microphysics, and showed the best agreement with in situ value (King et al. 2013). Furthermore, Zhang at al. (2012) showed that the 3.6 $\mu$m has shown is less sensitive to the plane-parallel cloud assumption.

It is explicitly mentioned (i.e. Page 1, Line 9-11; Page 3, Line 31-33) that the CDNC is not a product of MODIS and that the CDNC is identically derived from MODIS and COSP values of CER and COT following the methods of Bennartz et al. (2007), which is described in the dedicated Section 3.2. MODIS-derived CDNC is compared with the CDNC derived from COT and CER simulated from COSP as well as with the CDNC modeled direct output.

Regarding the spatial coverage of the CDNC bias distribution, the authors would prefer including both land and ocean. In order to clarify the possible implications of the assumptions embedded in the methodology used in the CDNC calculation, a new sentence has been added in the manuscript (see Specific Comment below). The discrepant results from the comparison of COSP-derived CDNC and the modeled direct output of CDNC, where the MODIS data represents solely the reference dataset rather than "truth" data, stem from the aerosol model set-up and this discrepancy has potentially important implications for the modelling community, using satellite observations to evaluate standard (not satellite-simulated) model output.

*The global CNDC-aerosol index sensitivity calculations (what the authors mistakenly call ACI throughout the manuscript) are also mostly descriptive and even though some speculation is given on the possible causes for discrepancy I imagine there is enough model output to go deeper (see specific comments). Also, in principle calculating the "ACI" from 2D vertically integrated fields makes little sense. It is not clear what role the assumptions of overlap are, or even whether the cloud and the aerosol occupy the same space.*

Author response:

We thank the Referee for addressing some well-known critical points in the satellite-based assessment of the ACI parameter. Some aspects are presented in the conclusion section (Page 19, Lines 10-30). However, in section 3.3 we added a paragraph that acknowledges the possible implications of the approach used to infer the CDNC-aerosol index sensitivity, or ACI.

The following paragraph is now added in Sect. 5:

"Some of the challenges and limitations in assessing ACI are now highlighted. MODIS AOD retrievals are limited to cloud-free conditions, which creates challenges to studying the ACI where the intention is to study collocated aerosol and cloud observations. Unless height-resolving instruments (i.e. lidars) are considered, the vertical location

**[ACPD](https://www.atmospheric-chemistry-and-physics-discussions.net/)**

Interactive
comment

of the aerosol is unknown. Aerosol and cloud measurements may contain retrieval errors, which are further propagated to ACI estimates, as well as they reciprocally may bias the respective retrievals (Jia et al., 2019). The interpretation of the observed aerosol-cloud relationships is further complicated by the effect of meteorology (Quaas et al., 2010; Gryspeerdt et al., 2014; Gryspeerdt et al., 2016; Brenguier2003). As cloud formation happens in high humidity conditions, aerosol humidification can severely affect the assessment of ACI by causing positive correlation between AOD and cloud properties (Myhre et al., 2007; Quaas et al.,2010; Gryspeerdt et al., 2014; Gryspeerdt et al., 2016). Additionally to aerosol particles, water vapour also affects precipitation (Boucher et al., 2013), obviously linked to the presence of clouds, and consequently causes spurious correlations between aerosols and clouds (Koren et al, 2012)."

The following additional references have been included in the manuscript:
Brenguier, J.-L., Pawlowska, H., and Schüller, L.: Cloud micro-physical and radiative properties for parameterization and satellite monitoring of the indirect effect of aerosol on climate, Journalof Geophysical Research: Atmospheres, 108, https://doi.org/10.1029/2002JD002682, URL-https://agupubs.onlinelibrary.wiley.com/doi/abs/10.1029/2002JD002682, 2003.
Gryspeerdt, E., Stier, P., and Grandey, B. S.: Cloud fraction mediates the aerosol optical depth-cloud top height relationship, Geo-physical Research Letters, 41, 3622–3627, https://doi.org/10.1002/2014GL059524, URL-https://agupubs.onlinelibrary.wiley.com/doi/abs/10.1002/2014GL059524, 2014.
Gryspeerdt, E., Quaas, J., and Bellouin, N.: Constraining the aerosol influence on cloud fraction, Journal of Geophysical Research: Atmo-spheres, 121, 3566–3583, https://doi.org/10.1002/2015JD023744, URL-https://agupubs.onlinelibrary.wiley.com/doi/abs/10.1002/2015JD023744, 2016.
Jia, H., Ma, X., Quaas, J., Yin, Y., and Qiu, T.: Is positive correlation be-tween cloud droplet effective radius and aerosol optical depth over land due to retrieval artifacts or real physical processes?, Atmospheric Chemistry andPhysics, 19,

8879–8896, https://doi.org/10.5194/acp-19-8879-2019, URLhttps://www.atmos-chem-phys.net/19/8879/2019/, 2019.

Koren, I., Oreopoulos, L., Feingold, G., Remer, L. A., and Altaratz, O.:Aerosol-induced intensification of rain from the tropics to the mid-latitudes,Nature Geoscience, 4, https://doi.org/10.1038/ngeo1364, URLhttps://doi.org/10.1038/ngeo1364, 2012.

Myhre, G., Stordal, F., Johnsrud, M., Kaufman, Y. J., Rosenfeld, D.,Storelvmo, T., Kristjansson, J. E., Berntsen, T. K., Myhre, A., and Isak-sen, I. S. A.: Aerosol-cloud interaction inferred from MODIS satellite data and global aerosol models, Atmospheric Chemistry and Physics, 7,3081–3101, https://doi.org/10.5194/acp-7-3081-2007, URLhttps://www.atmos-chem-phys.net/7/3081/2007/, 2007.

*Finally there is the issue of "the ACI". Aerosol-cloud interactions encompass many processes occurring in clouds as a result of the presence of aerosols. As a noun, aerosol cloud interactions is an area of study, not a metric. So it is troubling, and in many places grammatically incorrect, that ACI are reduced to a single number and equated to the CDNC-AI sensitivity. Please correct this and be precise in the terminology used.*

Author response:

We agree with the Reviewer that by definition the aerosol-cloud interactions encompasses different processes. The original text at Page 4, Lines 1-14, has been rephrase and expanded as:

"Aerosol-cloud interactions are based on the role of aerosol particles in changing cloud properties, which involves several processes (Bellouin et al., 2019). In this study we focus one of these processes, known as the first aerosol indirect effects or Twomey effect, which can be quantified by an indicator parameter, called ACI, defined as the change in an observable cloud property (e.g., cloud optical depth, cloud effective radius, cloud droplet number concentration) to a change in a cloud condensation nuclei proxy (e.g. aerosol optical depth, aerosol index, or aerosol particle number concentration)." The analysis of aerosol-cloud interactions has been

reported in literature by a variety of methods: studies presenting results from global scales (Feingold et al., 2001; Quaas et al., 2010) to regional scales(i.e Saponaro et al., 2017; BanWeiss et al., 2014; Liu et al., 2017; Liu et al., 2018) and in-situ observations (i.e. Sporre et al., 2014), using different approaches, i.e. observations from satellites, airborne and ground based instrumentation, or modelling. Nonetheless, the quantification of the aerosol-cloud interactions is still a major uncertainty in understanding climate change (eg. Lohmann et l., 2007; Quaas et al., 2009; Storelvmo et al., 2012; Boucher et al., 2013; Lee et al., 2015; Seinfeld et al., 2016; Bellouin et al., 2019).

The authors argue that the acronym ACI has already been used extensively in literature to purely define the mathematical relationship representing the sensitivity of clouds to changes in aerosol loading, eg. $ACI = dln(N_d)/dln(AI)$. The authors agree that using the acronym ACI for both the metric and the topic can be confusing for the readers, thus it is important to adopt one terminology for the mathematical expression and one for the more general discussions. Therefore, the manuscript is updated by using the term $ACI_{CDNC}$ only for the mathematical parameter representing the first aerosol indirect effect, as it common in literature, and the term "aerosol-cloud interactions" when the topic is discussed in general.

The following additional references have been included in the manuscript:
Bellouin, N. et al ( 2019). Bounding global aerosol radiative forcing of climate change. Reviews of Geophysics, 57. https://doi.org/10.1029/2019RG000660
Seinfeld J. H. et al. (2016). Improving our fundamental understanding of the role of aerosol-cloud interactions in the climate system. Proceedings of the National Academy of Sciences May 2016, 113 (21) 5781-5790; DOI: 10.1073/pnas.1514043113

*Specific comments:*

[Figure]

*Page 3, Line 34. Why 2008? The horizontal resolution is low enough that it should be easy to run a couple of years at least. MODIS data spans at least 15 years as well.*
Author response:
When the simulations were made, 2008 was the default year for AEROCOM model experiments. We also adopted this year to support the intercomparison of the results.

*Page 3, Line 35. No, the aerosol- cloud interaction is not a metric, and certainly not confused with the aerosol indirect effect. Even saying "the aerosol cloud interaction" is probably incorrect. Moreover, ACI is referred here as a metric and later on as a topic. Please be precise in the terms used.*
Author response:
The sentence has been rephrased as:
See General Comment above.

*Page 4 Line 32. This is the CDNC-AI sensitivity. Also I am not convinced that the meteorological component is completely removed, please explain.*
Author response: We do not claim that meteorologic influence are completely removed (Page 4, Line 21-23) and this is clearly stated in the added paragraph in Section 3.3 as well as in the conclusive section (Page 20, Lines 15-30).

*Page 4 Line 32. Why should the liquid water path remain constant? Is there a clear connection between the CDNC-AI sensitivity and the first aerosol indirect effect?*
Author response:
The aerosol-cloud interaction is a concept introduced to compare how changes in aerosol loading affect cloud properties. The ACI can be defined as:

$$ACI = \frac{d\ln COT}{d\ln(\sigma)} = -\frac{d\ln CER}{d\ln(\sigma)} = \frac{1}{3}\frac{d\ln CDNC}{d\ln(\sigma)} \tag{1}$$

where $\sigma$ is a proxy for the CCN concentration. From the theory (Twomey, 1977; Feingold et al., 2001; MCcomiskey and Feigold, 2008) for the first two parts of the equations, the LWP needs to be constant in the calculation of the partial derivatives while the third part of the equation (that is the one we use in our study) does not require any restriction of the LWP. Answering the Referee's second point: the CDNC-AI sensitivity is indeed an analytic approach to quantify the first aerosol indirect effect alternatively to the models described in the first two term of the Eq. (1).

*Page 9, Line 17. Why is the vertical resolution different than when running ECHAM-HAM?*
Author response:
The version of ECHAM-HAMMOZ with SALSA was the default configuration at the time when the simulations were run. However, it has to be noted that the difference between L31 of HAM and L47 of SALSA does not increase the vertical resolution of lowest grid-points but rather adds levels in the mid-atmosphere. The lowermost levels of L31 and L47 (up to about 100 hPa) are identical and Neubauer et al. (2019) have shown that the difference in cloud properties between the two vertical resolutions are minor.

*model calculation should be converted to in-cloud values before sending it to COSP to calculate MODIS-like values. Please clarify.*
Author response:
The calculation are correct. COSP does account for that MODIS only see in-cloud values. However, since the model output of cloud parameters from ESMs most often are grid-box averages, the in-cloud values from COSP are multiplied with cloud fraction before they are written as output. To get the in-cloud values one therefore needs to divide the output by cloud fraction in the post processing.

*Page 11, Line 18. Is there a MODIS algorithm to retrieve CDNC? It may be more correct to say in MODIS cloud effective radius is biased towards cloud top values, which may propagate to the CDNC calculation. The method used to estimate CDNC (equation 1) should approximate better cloud-base values.*
Author response:
See General Comment above.

*Page 12, Section 3.2. Please label this equation, and explain the assumptions behind it. Given than clouds are 3D, to what vertical level should this calculation correspond? Also, if this applies well in the stratiform marine boundary layer why are the authors applying it globally? What would be the error incurred in applying it to a shallow convective trade cumulus for example?*
Author response:
The authors thank the Referee for pointing out the absence of an explanation of the assumption behind the equation, now labelled in the manuscript as Eq.1. The following additional paragraph has been added in Sect.3.2.: "The assumption of not accounting for temperature effect and setting $\gamma$ as a bulk costant applies rather well to the warm stratiform clouds in the marine boundary layer but less for convective clouds (Bennartz, 2007; Rausch et al., 2010; Grosvenor et al, 2018). The equation represents the "Idealized Stratiform Boundary Layer Cloud" (ISBLC) model (Bennartz and Rausch, 2017) which is based on the following assumptions:

- the cloud is horizontally homogeneous;

- the LWC increases linearly from the cloud base to the cloud top;

- the CDNC is constant throughout the vertical extent of the cloud.

While the ISBLC model describes important aspects of stratiform boundary layer clouds, its assumption will never be fully valid for any real cloud. Issues related to the ISBLC model assumptions are extensively elaborated in (Bennartz, 2007; Bennartz and Rausch, 2017) and references therein."
The following reference was added to the references:
Grosvenor et al, (2018): Remote Sensing of Droplet Number Concentration in-Warm Clouds: A Review of the Current State of Knowledge and Per-spectives, Reviews of Geophysics, 56, 409–453, https://doi.org/10.1029/2017RG000593, URL-https://agupubs.onlinelibrary.wiley.com/doi/abs/10.1029/2017RG000593.

*Page 12, Line 15. This is the CDNC-AI sensitivity, not "the ACI".*
Author response:
Please, see Author response to the corresponding General Comment.

*Page 12, Lines 16-28. Please expand this. Is this done using a linear regression of the CDNC and AI daily time series? Given that this is a global calculation why would the number of data points be different? Also, it is not clear what the "ACI" from 2D vertically integrated fields represents (see general comments).*
Author response:
When calculating the ACI for the 1 x 1 degree grid points the ACI metric is applied

to data for a given season and grid box. This methodology can be thought of as calculating the linear regression slope of a scatter plot of ln(CDNC) vs. ln(AI), where each point represents a day for which both aerosol and cloud data exist for this grid box. As we follow directly the methodology presented by Grandey and Stier (2010), we do not repeat the details. The reader can find additional information in the mentioned reference. The number of data are different only when the dataset is divided accordinly to the season. A clarification regarding the ACI calculation is presented in the General Comment.

The following sentence was included in the manuscript:

"When calculating the $ACI_{CDNC}$ for the 1 degree x1 degree gridboxes Eq.1 is applied to data for the selected season and grid box. This methodology (Grandey and Stier, 2010) can be thought of as computing the linear regression slope of a scatter plot of ln(CDNC) versus ln(AI), where each point represents a day for which both aerosol and cloud data exist for the considered grid box."

*Page 13, Line 18. Does COSP produce COP CF or CF all? Maybe the discussion should be limited to that one.*
Author response:
COSP simulates $CF_{all}$. However, the author consider important to show also the results of COP-CF because it is valuable for modelling and satellite communities.

*Page 15, Lines 1-10. This is very vague and should be explored in more detail (see comments above).*
Author response:
Please, see Author response to the corresponding General Comment.

*Page 15, Lines 17-20. Is the difference due to the activation scheme or to the aerosol models?*

Author response:
The difference is due to the difference in aerosol models since they both use the same ICNC scheme.

*Page 15, Lines 25-30. These two sentences contradict each other.*
Author response:
The sentence has been rephrased as follows:
"Despite the higher CDNC, $CER_{liq}$ is larger in ECHAM-HAM-SALSA than in ECHAM-HAM. Although it is expected that $CER_{liq}$ decreases with increasing CDNC, higher LWP in ECHAM-HAM-SALSA in turn results in larger $CER_{liq}$ and outweighs the CDNC effect on $CER_{liq}$. The causes for differences in LWP between the two model versions are very difficult to diagnose but candidates for causing them are differences in wet scavenging, convective detrainment or freezing."

*Page 15, Lines 30-35. This is purely speculative.*
Author response:
The sentence has been deleted. Part of the content is included in the previous sentence. See previous comment.

*Page 17, Line 15-25. Could this be corroborated? It seems odd that the sensitivity would be negative.*
Author response:
Several studies, some of which we mention in the manuscript, have found negative values of ACI.

*Page 17, Line 25. Global maps of CDNC-AI sensitivity should show this better.*
Author response:

The Authors argue that a bar plot provides a more concise elaboration of the ACI estimates including the 95% confidence intervals. Additionally, we argue that the plot bar enables a quick comparison with similar studies. For these reasons, the authors wish to maintain Fig. 7 in its original plot.

**Supplement:**

[revised manuscript text omitted]